# Gated recurrence enables simple and accurate sequence prediction in stochastic, changing, and structured environments

**Cédric Foucault[1,2], Florent Meyniel[1]\***

[1]Cognitive Neuroimaging Unit, INSERM, CEA, Université Paris-Saclay, NeuroSpin center, Gif sur Yvette, France; [2]Sorbonne Université, Collège Doctoral, Paris, France

**Abstract** From decision making to perception to language, predicting what is coming next is crucial. It is also challenging in stochastic, changing, and structured environments; yet the brain makes accurate predictions in many situations. What computational architecture could enable this feat? Bayesian inference makes optimal predictions but is prohibitively difficult to compute. Here, we show that a specific recurrent neural network architecture enables simple and accurate solutions in several environments. This architecture relies on three mechanisms: gating, lateral connections, and recurrent weight training. Like the optimal solution and the human brain, such networks develop internal representations of their changing environment (including estimates of the environment's latent variables and the precision of these estimates), leverage multiple levels of latent structure, and adapt their effective learning rate to changes without changing their connection weights. Being ubiquitous in the brain, gated recurrence could therefore serve as a generic building block to predict in real-life environments.

**\*For correspondence:**
florent.meyniel@cea.fr

**Competing interest:** The authors declare that no competing interests exist.

## Editor's evaluation

There has been a longstanding interest in developing normative models of how humans handle latent information in stochastic and volatile environments. This study examines recurrent neural network models trained on sequence-prediction tasks analogous to those used in human cognitive studies. The results demonstrate that such models lead to highly accurate predictions for challenging sequences in which the statistics are non-stationary and change at random times. These novel and remarkable results open up new avenues for cognitive modelling.

## Introduction

Being able to correctly predict what is coming next is advantageous: it enables better decisions (*Dolan and Dayan, 2013*; *Sutton and Barto, 1998*), a more accurate perception of our world, and faster reactions (*de Lange et al., 2018*; *Dehaene et al., 2015*; *Saffran et al., 1996*; *Sherman et al., 2020*; *Summerfield and de Lange, 2014*). In many situations, predictions are informed by a sequence of past observations. In that case, the prediction process formally corresponds to a statistical inference that uses past observations to estimate latent variables of the environment (e.g. the probability of a stimulus) that then serve to predict what is likely to be observed next. Specific features of real-life environments make this inference a challenge: they are often partly random, changing, and structured in different ways. Yet, in many situations, the brain is able to overcome these challenges and shows several aspects of the optimal solution (*Dehaene et al., 2015*; *Dolan and Dayan, 2013*; *Gallistel*

*et al., 2014*; *Summerfield and de Lange, 2014*). Here, we aim to identify the computational mechanisms that could enable the brain to exhibit these aspects of optimality in these environments.

We start by unpacking two specific challenges which arise in real-life environments. First, the joint presence of randomness and changes (i.e. the non-stationarity of the stochastic process generating the observations) poses a well-known tension between stability and flexibility (*Behrens et al., 2007*; *Soltani and Izquierdo, 2019*; *Sutton, 1992*). Randomness in observations requires integrating information over time to derive a stable estimate. However, when a change in the estimated variable is suspected, it is better to limit the integration of past observations to update the estimate more quickly. The prediction should thus be adaptive, that is, dynamically adjusted to promote flexibility in the face of changes and stability otherwise. Past studies have shown that the brain does so in many contexts: perception (*Fairhall et al., 2001*; *Wark et al., 2009*), homeostatic regulation (*Pezzulo et al., 2015*; *Sterling, 2004*), sensorimotor control (*Berniker and Kording, 2008*; *Wolpert et al., 1995*), and reinforcement learning (*Behrens et al., 2007*; *Iglesias et al., 2013*; *Soltani and Izquierdo, 2019*; *Sutton and Barto, 1998*).

Second, the structure of our environment can involve complex relationships. For instance, the sentence beginnings "what science can do for you is..." and "what you can do for science is..." call for different endings even though they contain the same words, illustrating that prediction takes into account the ordering of observations. Such structures appear not only in human language but also in animal communication (*Dehaene et al., 2015*; *Hauser et al., 2001*; *Robinson, 1979*; *Rose et al., 2004*), and all kinds of stimulus-stimulus and stimulus-action associations in the world (*Saffran et al., 1996*; *Schapiro et al., 2013*; *Soltani and Izquierdo, 2019*; *Sutton and Barto, 1998*). Such a structure is often latent (i.e. not directly observable) and it governs the relationship between observations (e.g. words forming a sentence, stimulus-action associations). These relationships must be leveraged by the prediction, making it more difficult to compute.

In sum, the randomness, changes, and latent structure of real-life environments pose two major challenges: that of adapting to changes and that of leveraging the latent structure. Two commonly used approaches offer different solutions to these challenges. The Bayesian approach allows to derive statistically optimal predictions for a given environment knowing its underlying generative model. This optimal solution is a useful benchmark and has some descriptive validity since, in some contexts, organisms behave close to optimally (*Ma and Jazayeri, 2014*; *Tauber et al., 2017*) or exhibit several qualitative aspects of the optimal solution (*Behrens et al., 2007*; *Heilbron and Meyniel, 2019*; *Meyniel et al., 2015*). However, a specific Bayes-optimal solution only applies to a specific generative model (or class of models [*Tenenbaum et al., 2011*]). This mathematical solution also does not in general lead to an algorithm of reasonable complexity (*Cooper, 1990*; *Dagum and Luby, 1993*). Bayesian inference therefore says little about the algorithms that the brain could use, and the biological basis of those computations remains mostly unknown with only a few proposals highly debated (*Fiser et al., 2010*; *Ma et al., 2006*; *Sahani and Dayan, 2003*).

Opposite to the Bayes-optimal approach is the heuristics approach: solutions that are easy to compute and accurate in specific environments (*Todd and Gigerenzer, 2000*). However, heuristics lack generality: their performance can be quite poor outside the environment that suits them. In addition, although simple, their biological implementation often remains unknown (besides the delta-rule [*Eshel et al., 2013*; *Rescorla and Wagner, 1972*; *Schultz et al., 1997*]).

Those two approaches leave open the following questions: Is there a general, biologically feasible architecture that enables, in different environments, solutions that are simple, effective, and that reproduce the qualitative aspects of optimal prediction observed in organisms? If so, what are its essential mechanistic elements?

Our approach stands in contrast with the elegant closed-form but intractable mathematical solutions offered by Bayesian inference, and the simple but specialized algorithms offered by heuristics. Instead, we look for general mechanisms under the constraints of feasibility and simplicity. We used recurrent neural networks because they can offer a generic, biologically feasible architecture able to realize different prediction algorithms (see *LeCun et al., 2015*; *Saxe et al., 2021* and Discussion). We used small network sizes in order to produce simple (i.e. low-complexity, memory-bounded) solutions. We tested their generality using different environments. To determine the simplest architecture sufficient for effective solutions and derive mechanistic insights, we considered different architectures that varied in size and mechanisms. For each one, we instantiated several networks and

trained them to approach their best possible prediction algorithm in a given environment. We treated the training procedure as a methodological step without claiming it to be biologically plausible. To provide interpretability, we inspected the networks' internal model and representations, and tested specific optimal aspects of their behavior—previously reported in humans (*Heilbron and Meyniel, 2019*; *Meyniel et al., 2015*; *Nassar et al., 2010*; *Nassar et al., 2012*)—which demonstrate the ability to adapt to changes and leverage the latent structure of the environment.

## Results

### The framework: sequence prediction and network architectures

All our analyses confront simulated agents with the same general problem: sequence prediction. It consists in predicting, at each time step in a sequence where one time step represents one observation, the probability distribution over the value of the next observation given the previous observations (here we used binary observations coded as '0' and '1') (*Figure 1a*). The environment generates the sequence, and the agent's goal is to make the most accurate predictions possible in this environment. Below, we introduce three environments. All of them are stochastic (observations are governed by latent probabilities) and changing (these latent probabilities change across time), and thus require dynamically adapting the stability-flexibility tradeoff. They also feature increasing levels of latent structure that must be leveraged, making the computation of predictions more complex.

How do agents learn to make predictions that fit a particular environment? In real life, agents often do not benefit from any external supervision and must rely only on the observations. To do so, they can take advantage of an intrinsic error signal that measures the discrepancy between their prediction and the actual value observed at the next time step. We adopted this learning paradigm (often called unsupervised, self-supervised, or predictive learning in machine learning [*Elman, 1991*; *LeCun, 2016*]) to train our agents in silico. We trained the agents by exposing them to sequences generated by a given environment and letting them adjust their parameters to improve their prediction (see Materials and methods).

During testing, we kept the parameters of the trained agents frozen, exposed them to new sequences, and performed targeted analyses to probe whether they exhibit specific capabilities and better understand how they solve the problem.

Our investigation focuses on a particular class of agent architectures known as recurrent neural networks. These are well suited for sequence prediction because recurrence allows to process inputs sequentially while carrying information over time in recurrent activity. The network architectures we used all followed the same three-layer template, consisting of one input unit whose activity codes for the current observation, one output unit whose activity codes for the prediction about the next observation, and a number of recurrent units that are fed by the input unit and project to the output unit (*Figure 1b*). All architectures had self-recurrent connections.

We identified three mechanisms of recurrent neural network architectures that endow a network with specific computational properties which have proven advantageous in our environments (*Figure 1c*). One mechanism is gating, which allows for multiplicative interactions between the activities of units. A second mechanism is lateral connectivity, which allows the activities of different recurrent units to interact with each other. A third mechanism is the training of recurrent connection weights, which allows the dynamics of recurrent activities to be adjusted to the training environment.

To get mechanistic insight, we compared an architecture that included all three mechanisms, to alternative architectures that were deprived of one of the three mechanisms but maintained the other two (*Figure 1d*; see Materials and methods for equations). Here, we call an architecture with all three mechanisms 'gated recurrent', and the particular architecture we used is known as GRU (*Cho et al., 2014*; *Chung et al., 2014*). When deprived of gating, multiplicative interactions between activities are removed, and the architecture reduces to that of a vanilla recurrent neural network also known as the Elman network (*Elman, 1990*). When deprived of lateral connections, the recurrent units become independent of each other, thus each recurrent unit acts as a temporal filter on the input observations (with possibly time-varying filter weights thanks to gating). When deprived of recurrent weight training, the recurrent activity dynamics become independent of the environment and the only parameters that can be trained are those of the output unit; this architecture is thus one form of reservoir computing (*Tanaka et al., 2019*). In the results below, unless otherwise stated, the networks all had

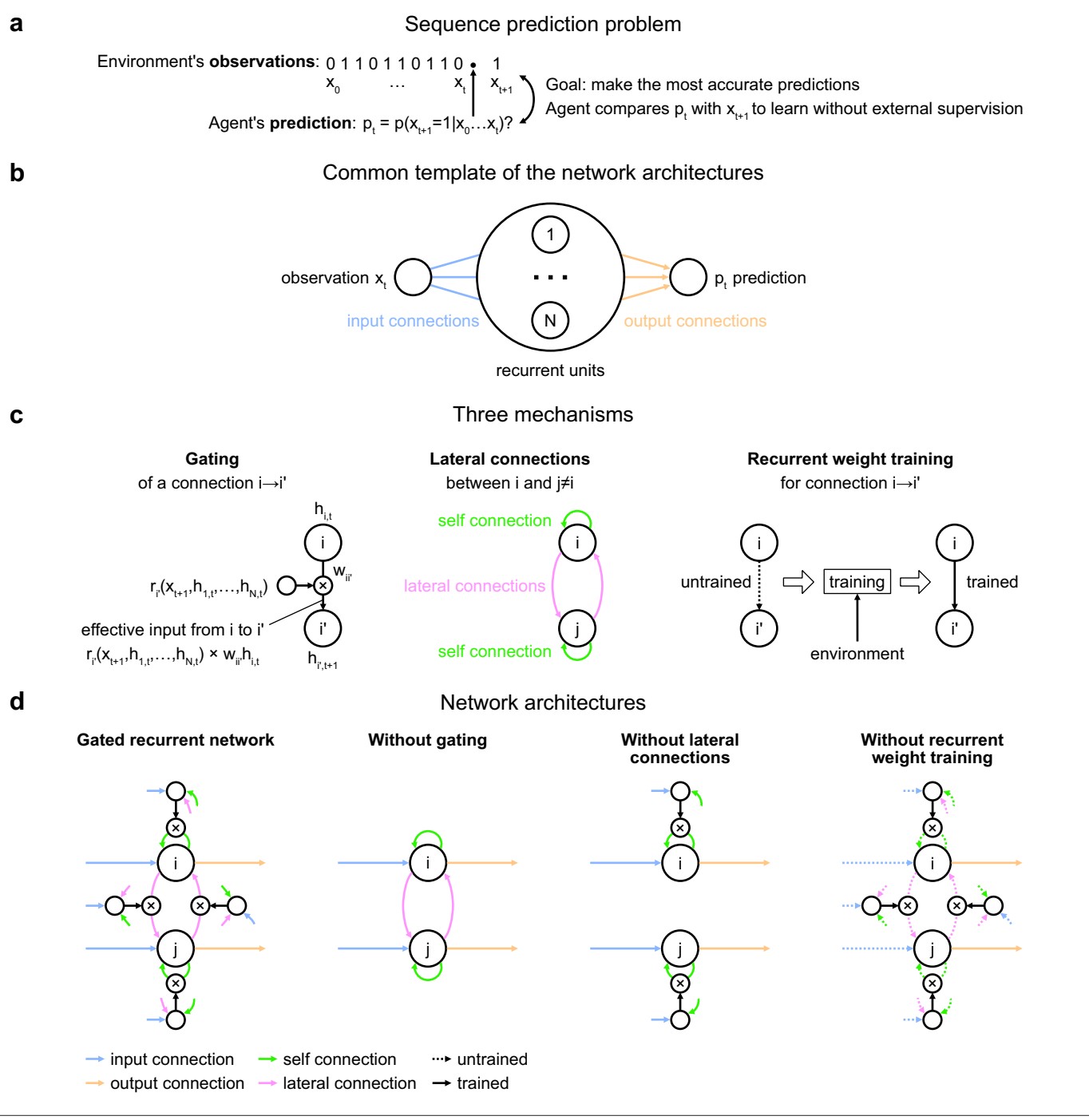

**Figure 1.** Problem to solve and network architectures. (**a**) Sequence prediction problem. At each time step t, the environment generates one binary observation $x_t$. The agent receives it and returns a prediction $p_t$: its estimate of the probability that the next observation will be one given the observations collected so far. The agent's goal is to make the most accurate predictions possible. The agent can measure its accuracy by comparing its prediction $p_t$ with the actual value observed at the next time step $x_{t+1}$, allowing it to learn from the observations without any external supervision. (**b**) Common three-layer template of the recurrent neural network architectures. Input connections transmit the observation to the recurrent units and output connections allow the prediction to be read from the recurrent units. (**c**) Three key mechanisms of recurrent neural network architectures. Gating allows for multiplicative interaction between activities. Lateral connections allow the activities of different recurrent units i and j to interact. Recurrent weight training allows the connection weights of recurrent units to be adjusted to the training environment. i' may be equal to i. (**d**) The gated recurrent architecture includes all three mechanisms: gating, lateral connections, and recurrent weight training. Each alternative architecture includes all but one of the three mechanisms.

The online version of this article includes the following figure supplement(s) for figure 1:

*Figure 1 continued on next page*

*Figure 1 continued*

**Figure supplement 1.** Graphical model of the generative process of each environment.

11 recurrent units (the smallest network size beyond which the gated recurrent network showed no substantial increase in performance in any of the environments), but the results across architectures are robust to this choice of network size (see the last section of the Results).

## Performance in the face of changes in latent probabilities

We designed a first environment to investigate the ability to handle changes in a latent probability (*Figure 2a*; see *Figure 1—figure supplement 1* for a graphical model). In this environment we used the simplest kind of latent probability: p(1), the probability of occurrence (or base rate) of the observation being 1 (note that p(0) = 1−p(1)), here called 'unigram probability'. The unigram probability suddenly changed from one value to another at so-called 'change points', which could occur at any time, randomly with a given fixed probability.

This environment, here called 'changing unigram environment', corresponds for instance to a simple oddball task (*Aston-Jones et al., 1997*; *Kaliukhovich and Vogels, 2014*; *Ulanovsky et al., 2004*), or the probabilistic delivery of a reward with abrupt changes in reward probabilities (*Behrens et al., 2007*; *Vinckier et al., 2016*). In such an environment, predicting accurately is difficult due to the stability-flexibility tradeoff induced by the stochastic nature of the observations (governed by the unigram probability) and the possibility of a change point at any moment.

To assess the networks' prediction accuracy, we compared the networks with the optimal agent for this specific environment, that is, the optimal solution to the prediction problem determined using Bayesian inference. This optimal solution knows the environment's underlying generative process and uses it to compute, via Bayes' rule, the probability distribution over the possible values of the latent probability given the past observation sequence, $p(p_{t+1}^{env}|x_0, ..., x_t)$ known as the posterior distribution. It then outputs as prediction the mean of this distribution. (For details see Materials and methods and *Heilbron and Meyniel, 2019*).

We also compared the networks to two types of heuristics which perform very well in this environment: the classic 'delta-rule' heuristic (*Rescorla and Wagner, 1972*; *Sutton and Barto, 1998*) and the more accurate 'leaky' heuristic (*Gijsen et al., 2021*; *Heilbron and Meyniel, 2019*; *Meyniel et al.,*

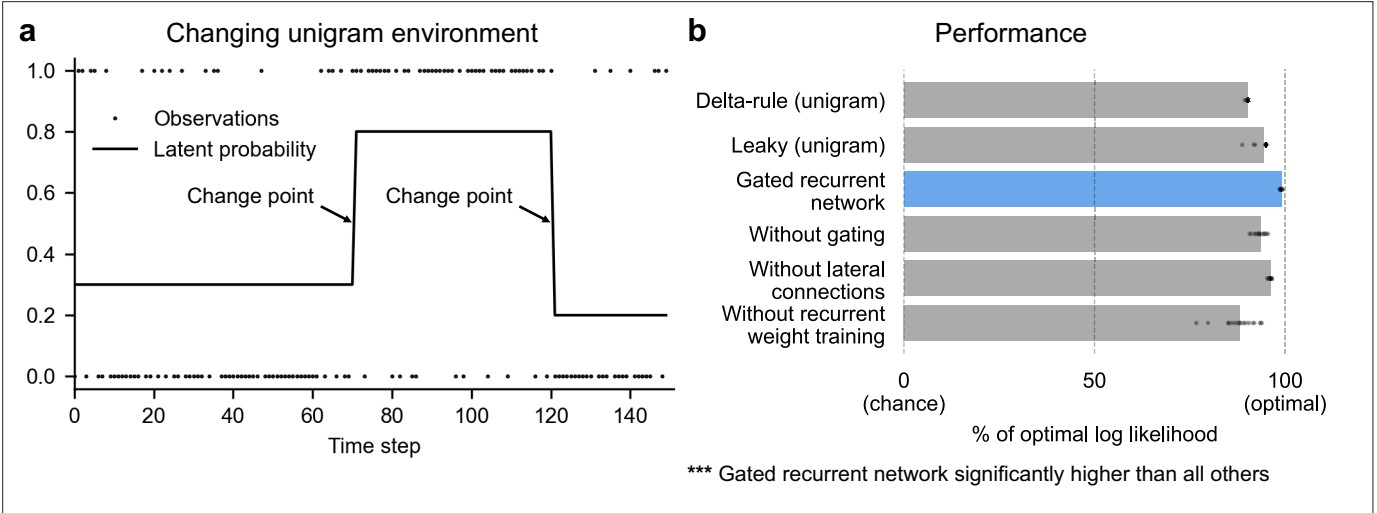

**Figure 2.** Gated recurrent networks perform quasi-optimally in the face of changes in latent probabilities. (**a**) Sample sequence of observations (dots) and latent unigram probability (line) generated in the changing unigram environment. At each time step, a binary observation is randomly generated based on the latent unigram probability, and a change point can occur with a fixed probability, suddenly changing the unigram probability to a new value uniformly drawn in [0,1]. (**b**) Prediction performance in the changing unigram environment. For each type of agent, 20 trained agents (trained with different random seeds) were tested (dots: agents; bars: average). Their prediction performance was measured as the % of optimal log likelihood (0% being chance performance and 100% optimal performance, see *Equation 1* for the log likelihood) and averaged over observations and sequences. The gated recurrent network significantly outperformed every other type of agent (p < 0.001, two-tailed two independent samples t-test with Welch's correction for unequal variances).

*2016*; *Yu and Cohen, 2008*) (see Materials and methods for details). To test the statistical reliability of our conclusions, we trained separately 20 agents of each type (each type of network and each type of heuristic).

We found that even with as few as 11 units, the gated recurrent networks performed quasi-optimally. Their prediction performance was 99% of optimal (CI±0.1%), 0% corresponding to chance level (*Figure 2b*). Being only 1% short of optimal, the gated recurrent networks outperformed the delta rule and leaky agents, which performed 10 times and 5 times further from optimal, respectively (*Figure 2b*).

For mechanistic insight, we tested the alternative architectures deprived of one mechanism. Without either gating, lateral connections, or recurrent weight training, the average performance was respectively 6 times, 4 times, and 12 times further from optimal (*Figure 2b*), that is, the level of a leaky agent or worse. The drops in performance remain similar when considering only the best network of each architecture instead of the average performance (*Figure 2b*, compare rightmost dots across rows).

These results show that small gated recurrent networks can achieve quasi-optimal predictions and that the removal of one of the mechanisms of the gated recurrent architecture results in a systematic drop in performance.

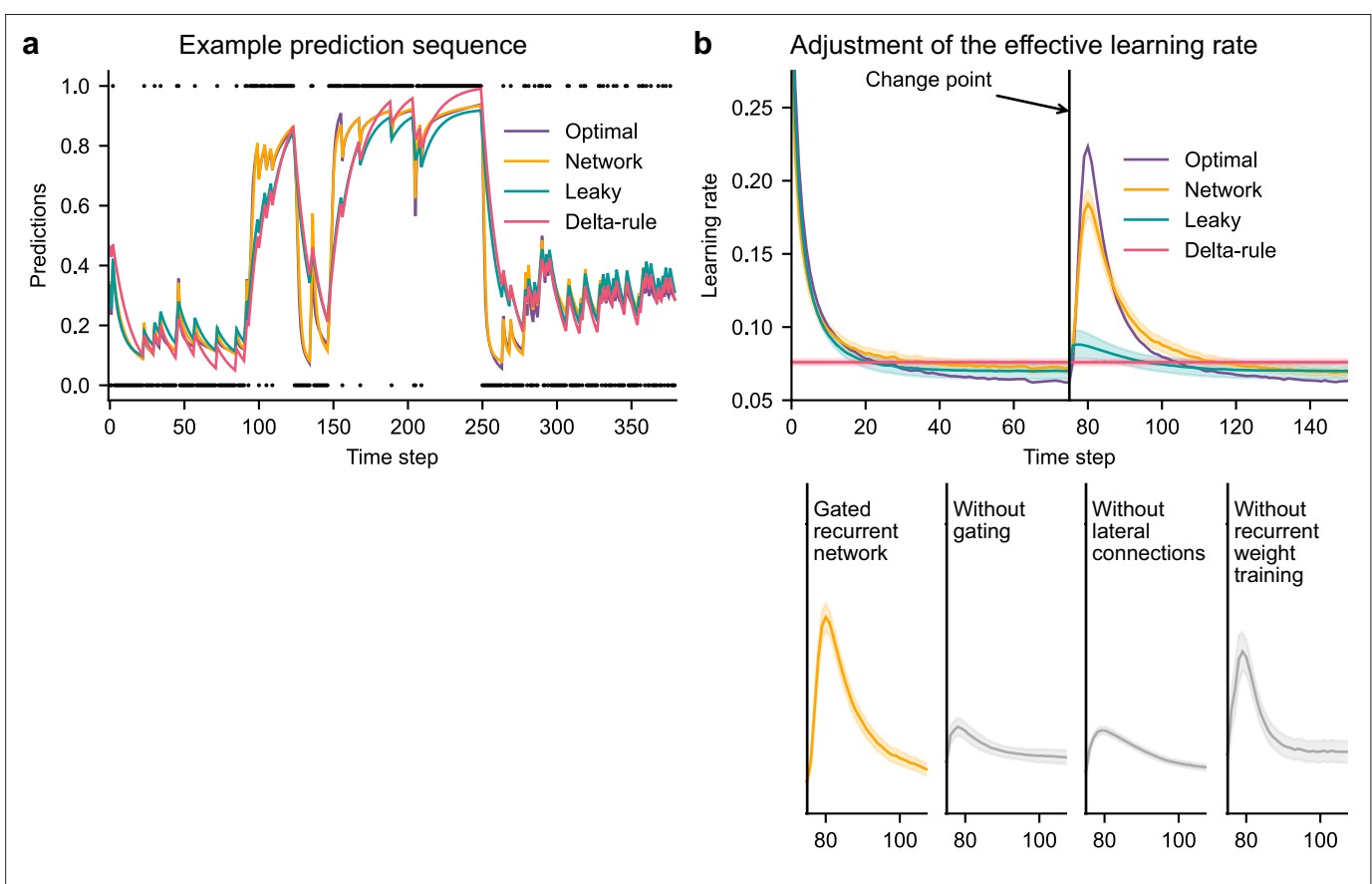

**Figure 3.** Gated recurrent but not alternative networks adjust their moment-by-moment effective learning rate around changes like the optimal agent. (**a**) Example prediction sequence illustrating the prediction updates of different types of agents. Within each type of agent, the agent (out of 20) yielding median performance in *Figure 2b* was selected for illustration purposes. Dots are observations, lines are predictions. (**b**) Moment-by-moment effective learning rate of each type of agent. 20 trained agents of each type were tested on 10,000 sequences whose change points were locked at the same time steps, for illustration purposes. The moment-by-moment effective learning rate was measured as the ratio of prediction update to prediction error (see Materials and methods, *Equation 2*), and averaged over sequences. Lines and bands show the mean and the 95% confidence interval of the mean.

The online version of this article includes the following figure supplement(s) for figure 3:

**Figure supplement 1.** Attunement of the effective learning rate to the change point probabilities.

## Adaptation to changes through the adjustment of the effective learning rate

In a changing environment, the ability to adapt to changes is key. Networks exposed to more changing environments during training updated their predictions more overall during testing, similarly to the optimal agent (see *Figure 3—figure supplement 1*) and, to some extent, humans (*Behrens et al., 2007*, Figure 2e; *Findling et al., 2021*, Figure 4c). At a finer timescale, the moment-by-moment updating of the predictions also showed sensible dynamics around change points.

*Figure 3a* illustrates a key difference in behavior between, on the one hand, the optimal agent and the gated recurrent network, and on the other hand, the heuristic agents: the dynamics of their update differ. This difference is particularly noticeable when recent observations suggest that a change point has just occurred: the optimal agent quickly updates the prediction by giving more weight to the new observations; the gated recurrent network behaves the same but not the heuristic agents. We formally tested this dynamic updating around change points by measuring the moment-by-moment effective learning rate, which normalizes the amount of update in the prediction by the prediction error (i.e. the difference between the previous prediction and the actual observation; see Materials and methods, *Equation 2*).

Gated recurrent networks turned out to adjust their moment-by-moment effective learning rate as the optimal agent did, showing the same characteristic peaks, at the same time and with almost the same amplitude (*Figure 3b*, top plot). By contrast, the effective learning rate of the delta-rule agents was (by construction) constant, and that of the leaky agents changed only marginally.

When one of the mechanisms of the gated recurrence was taken out, the networks' ability to adjust their effective learning rate was greatly degraded (but not entirely removed) (*Figure 3b*, bottom plots). Without gating, without lateral connections, or without recurrent weight training, the amplitude was lower (showing both a lower peak value and a higher baseline value), and the peak occurred earlier.

This shows that gated recurrent networks can reproduce a key aspect of optimal behavior: the ability to adapt the update of their prediction to change points, which is lacking in heuristic agents and alternative networks.

## Internal representation of precision and dynamic interaction with the prediction

Beyond behavior, we sought to determine whether a network's ability to adapt to changes relied on idiosyncratic computations or followed the more general principle of precision-weighting derived from probability theory. According to this principle, the precision of the current prediction (calculated in the optimal agent as the negative logarithm of the standard deviation of the posterior distribution over the latent probability, see *Equation 3* in Materials and methods) should influence the weight of the current prediction relative to the next observation in the updating process: for a given prediction error, the lower the precision, the higher the subsequent effective learning rate. This precision-weighting principle results in an automatic adjustment of the effective learning rate in response to a change, because the precision of the prediction decreases when a change is suspected.

In line with this principle, human participants can estimate not only the prediction but also its precision as estimated by the optimal agent (*Boldt et al., 2019*, Figure 2; *Meyniel et al., 2015*, Figure 4B), and this precision indeed relates to the participants' effective learning rate (*McGuire et al., 2014*, Figure 2C and S1A; *Nassar et al., 2010*, Figure 4C and 3B; *Nassar et al., 2012*, Figure 5 and 7c, ).

We tested whether a network could represent this optimal precision too, by trying to linearly read it from the network's recurrent activity (*Figure 4a*). Note that the networks were trained only to maximize prediction accuracy (not to estimate precision). Yet, in gated recurrent networks, we found that the read precision on left-out data was highly accurate (*Figure 4a*, left plot: the median Pearson correlation with the optimal precision is 0.82), and correlated with their subsequent effective learning rate as in the optimal agent (*Figure 4a*, right plot: the median correlation for gated recurrent networks is –0.79; for comparison, it is –0.88 for the optimal agent).

To better understand how precision information is represented and how it interacts with the prediction dynamically in the network activity, we plotted the dynamics of the network activity in the subspace spanned by the prediction and precision vectors (*Figure 4b*). Such visualization captures both the temporal dynamics and the relationships between the variables represented in the network,

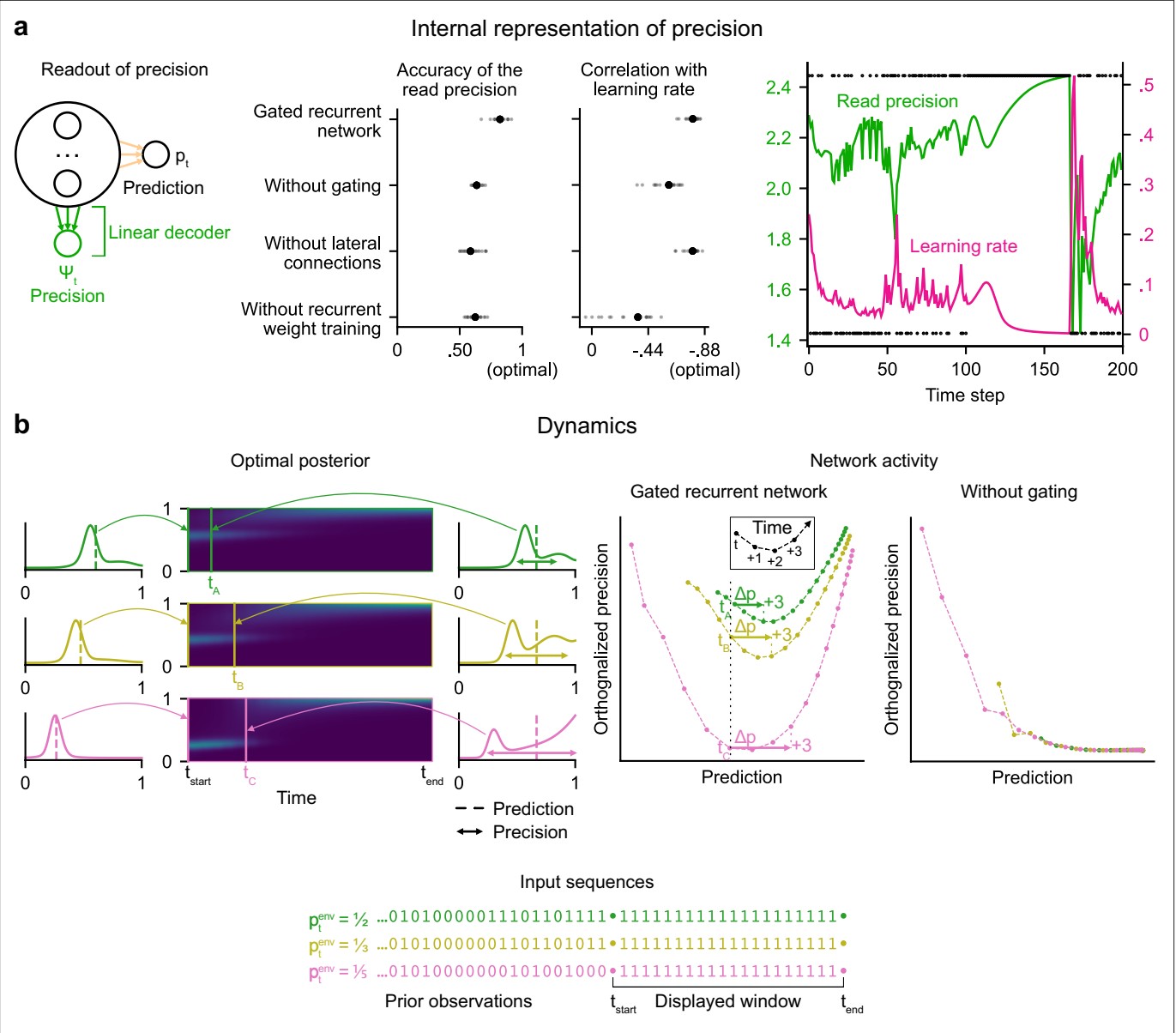

**Figure 4.** Gated recurrent networks have an internal representation of the precision of their estimate that dynamically interacts with the prediction following the precision-weighting principle. (**a**) Left to right: Schematic of the readout of precision from the recurrent activity of a network (obtained by fitting a multiple linear regression from the recurrent activity to the log precision of the optimal posterior distribution); Accuracy of the read precision (calculated as its Pearson correlation with the optimal precision); Pearson correlation between the read precision and the network's subsequent effective learning rate (the optimal value was calculated from the optimal agent's own precision and learning rate); Example sequence illustrating their anti-correlation in the gated recurrent network. In both dot plots, large and small dots show the median and individual values, respectively. (**b**) Dynamics of the optimal posterior (left) and the network activity (right) in three sequences (green, yellow, and pink). The displayed dynamics are responses to a streak of 1 s after different sequences of observations (with different generative probabilities as shown at the bottom). The optimal posterior distribution is plotted as a color map over time (dark blue and light green correspond to low and high probability densities, respectively) and as a line plot at two times: on the left, the time $t_{start}$ just before the streak of 1s, and on the right, a time $t_A/t_B/t_C$ when the prediction (i.e. mean) is approximately equal in all three cases; note that the precision differs. The network activity was projected onto the two-dimensional subspace spanned by the prediction and precision vectors (for the visualization, the precision axis was orthogonalized with respect to the prediction axis). In the gated recurrent network, the arrow $\Delta p$ shows the update to the prediction performed in the next three time steps starting at the time $t_A/t_B/t_C$ defined from the optimal posterior. Like the optimal posterior and unlike the network without gating, the gated recurrent network represents different levels of precision at an equal prediction, and the lower the precision, the higher the subsequent update to the prediction—a principle called precision-weighting. In all example plots (**a–b**), the displayed network is the one of the 20 that yielded the median read precision accuracy.

and has helped understand network computations in other works (*Mante et al., 2013*; *Sohn et al., 2019*). Here, two observations can be made.

First, in the gated recurrent network (*Figure 4b*, second plot from the right), the trajectories are well separated along the precision axis (for the same prediction, the network can represent multiple precisions), meaning that the representation of precision is not reducible to the prediction. By contrast, in the network without gating (*Figure 4b*, rightmost plot), these trajectories highly overlap, which indicates that the representation of precision and prediction are mutually dependent. To measure this dependence, we computed the mutual information between the read precision and the prediction of the network, and it turned out to be very high in the network without gating (median MI = 5.2) compared to the gated recurrent network (median MI = 0.7) and the optimal agent (median MI = 0.6) (without lateral connections, median MI = 1.3; without recurrent weight training, median MI = 1.9), confirming that gating is important to separate the precision from the prediction.

Second, in the gated recurrent network, the precision interacts dynamically with the prediction in a manner consistent with the precision-weighting principle: for a given prediction, the lower the precision, the larger the subsequent updates to the prediction (*Figure 4b*, vertical dotted line indicates the level of prediction and arrows the subsequent updates).

These results indicate that in the network without gating, precision is confounded with prediction and the correlation between precision and effective learning rate is spuriously driven by the prediction itself, whereas in the network with gating, there is a genuine representation of precision beyond the prediction itself, which interacts with the updating of predictions. However, we have so far only provided correlational evidence; to show that the precision represented in the network plays a causal role in the subsequent prediction update, we need to perform an intervention that acts selectively on this precision.

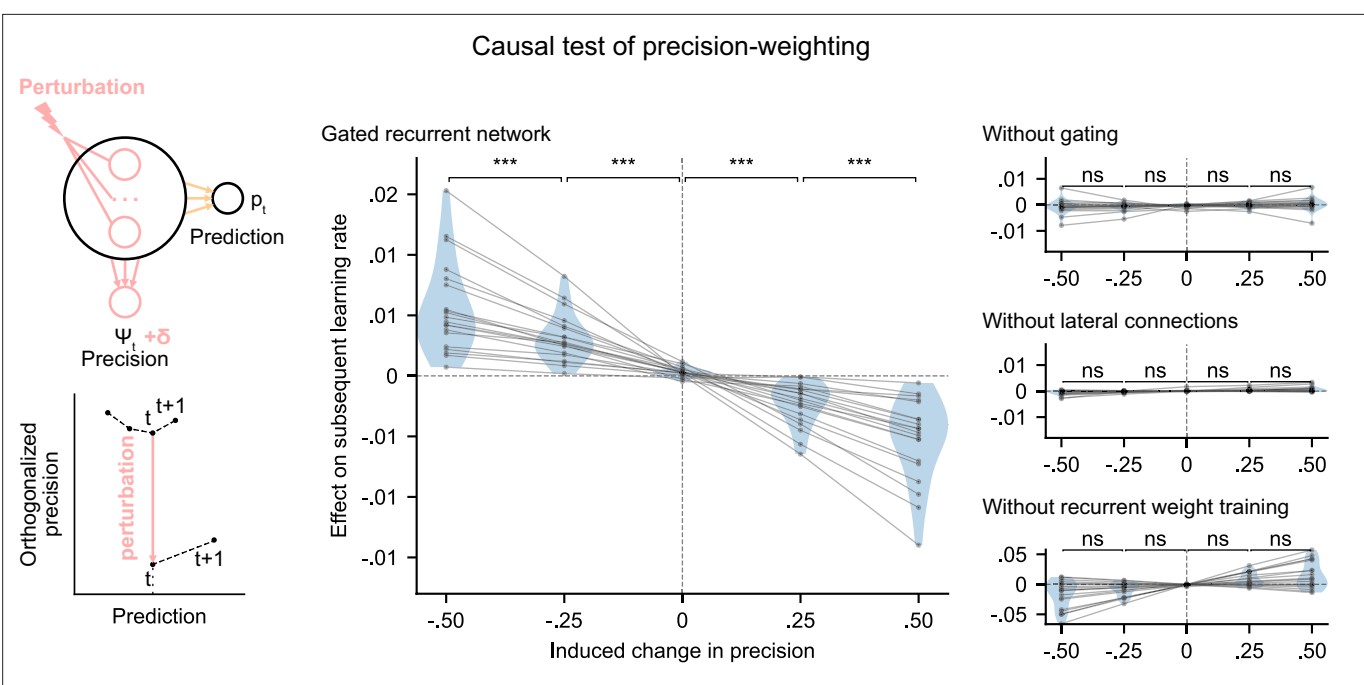

**Figure 5.** Precision-weighting causally determines the adjustment of the effective learning rate in gated recurrent networks only. Causal test of a network's precision on its effective learning rate. The recurrent activity was perturbed to induce a controlled change δ in the read precision, while keeping the prediction at the current time step—and thus the prediction error at the next time step—constant. This was done by making the perturbation vector orthogonal to the prediction vector and making its projection onto the precision vector equal to δ (bottom left diagram). We measured the perturbation's effect on the subsequent effective learning rate as the difference in learning rate 'with perturbation' minus 'without perturbation' at the next time step (four plots on the right). Each dot (and joining line) corresponds to one network. ***: p < 0.001, n.s.: p > 0.05 (one-tailed paired t-test).

## Causal role of precision-weighting for adaptation to changes

We tested whether the internal representation of precision causally regulated the effective learning rate in the networks using a perturbation experiment. We designed perturbations of the recurrent activity that induced a controlled change in the read precision, while leaving the networks' current prediction unchanged to control for the effect of the prediction error (for the construction of the perturbations, see *Figure 5* bottom left diagram and legend, and Materials and methods). These perturbations caused significant changes in the networks' subsequent effective learning rate, commensurate with the induced change in precision, as predicted by the principle of precision-weighting (*Figure 5*, middle plot). Importantly, this causal relationship was abolished in the alternative networks that lacked one of the mechanisms of the gated recurrent architecture (*Figure 5*, right three plots; the slope of the effect was significantly different between the gated recurrent network group and any of the alternative network groups, two-tailed two independent samples t-test, all $t(38) > 4.1$, all $p < 0.001$, all Cohen's $d > 1.3$).

These results show that the gated recurrent networks' ability to adapt to changes indeed relies on their precision-dependent updating and that such precision-weighting does not arise without all three mechanisms of the gated recurrence.

## Leveraging and internalizing a latent structure: bigram probabilities

While the changing unigram environment already covers many tasks in the behavioral and neuroscience literature, real-world sequences often exhibit more structure. To study the ability to leverage such structure, we designed a new stochastic and changing environment in which the sequence of observations is no longer generated according to a single unigram probability, $p(1)$, but two 'bigram probabilities' (also known as transition probabilities), $p(0|0)$ and $p(1|1)$, which denote the probability of occurrence of a 0 after a 0 and of a 1 after a 1, respectively (*Figure 6a*; see *Figure 1—figure supplement 1* for a graphical model). These bigram probabilities are also changing randomly, with independent change points.

This 'changing bigram environment' is well motivated because there is ample evidence that bigram probabilities play a key role in sequence knowledge in humans and other animals (*Dehaene et al., 2015*) even in the face of changes (*Bornstein and Daw, 2013*; *Meyniel et al., 2015*).

We assessed how well the networks could leverage the latent bigram structure after having been trained in this environment. For comparison, we tested the optimal agent for this environment as well as two groups of heuristics: delta-rule and leaky estimation of unigram probabilities (as in *Figure 2b*), and now also delta rule and leaky estimation of bigram probabilities (see Materials and methods for details).

The gated recurrent networks achieved 98% of optimal prediction performance (CI ±0.3%), outperforming the heuristic agents estimating bigram probabilities, and even more so those estimating a unigram probability (*Figure 6c*). To demonstrate that this was due to their internalization of the latent structure, we also tested the gated recurrent networks that had been trained in the changing unigram environment: their performance was much worse (*Figure 6—figure supplement 1*).

At the mechanistic level, all three mechanisms of the gated recurrence are important for this ability to leverage the latent bigram structure. Not only does the performance drop when one of these mechanisms is removed (*Figure 6c*), but also this drop in performance is much larger than that observed in the changing unigram environment (without gating: –11.2% [CI ±1.5% calculated by Welch's t-interval] in the bigram environment vs. –5.5% [CI ±0.6%] in the unigram environment; without lateral connections: –18.5% [CI ±1.8%] vs. –2.9% [CI ±0.2%]; without recurrent weight training: –29.9% [CI ±1.6%] vs. –11.0% [CI ±2.1%]; for every mechanism, there was a significant interaction effect between the removal of the mechanism and the environment on performance, all $F(1,76) > 47.9$, all $p < 0.001$).

*Figure 6b* illustrates the gated recurrent networks' ability to correctly incorporate the bigram context into its predictions compared to networks lacking one of the mechanisms of the gated recurrence. While a gated recurrent network aptly changes its prediction from one observation to the next according to the preceding observation as the optimal agent does, the other networks fail to show such context-dependent behavior, sometimes even changing their prediction away from the optimal agent.

Altogether these results show that gated recurrent networks can leverage the latent bigram structure, but this ability is impaired when one mechanism of the gated recurrence is missing.

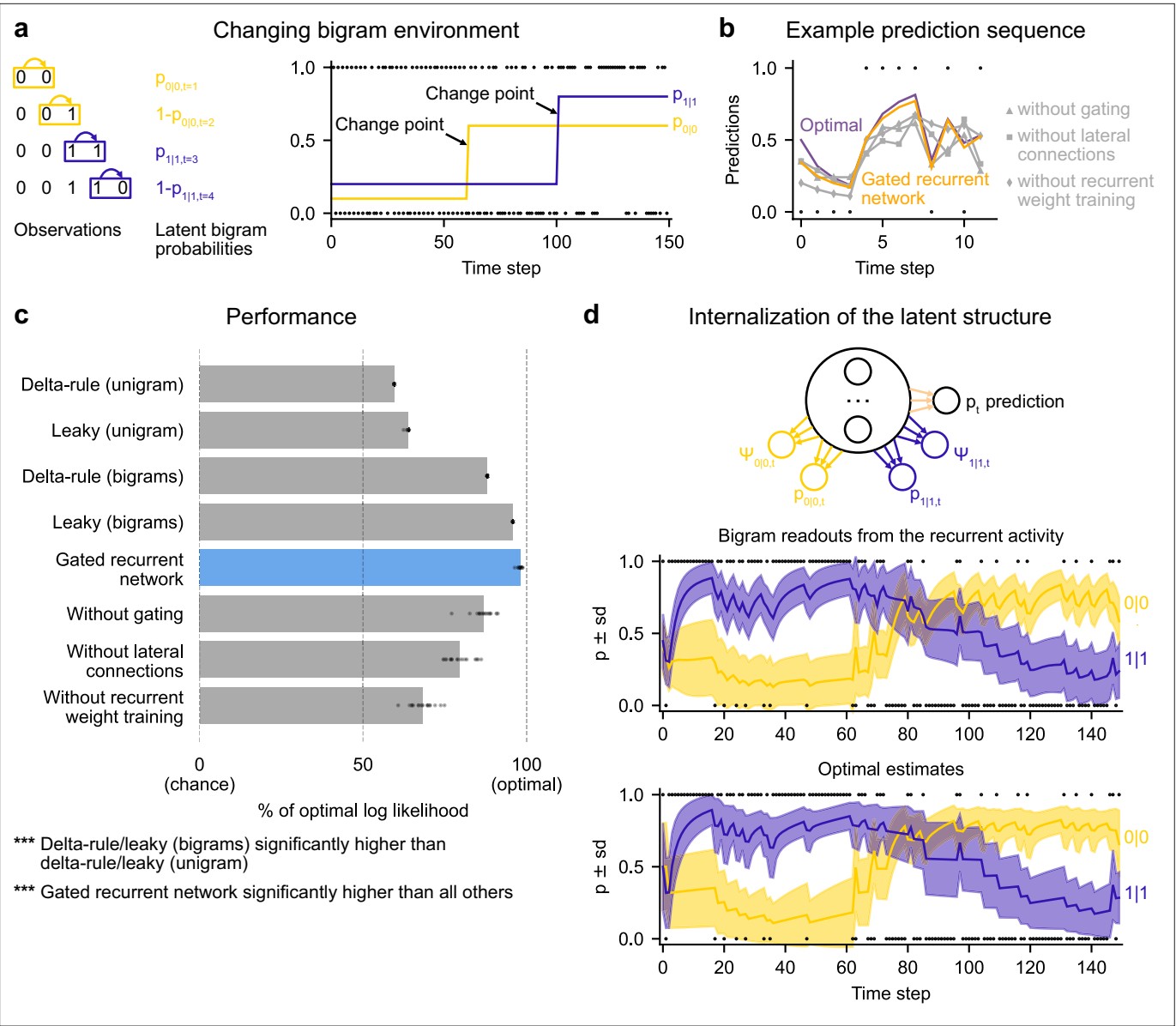

**Figure 6.** Gated recurrent networks correctly leverage and internalize the latent bigram structure. (**a**) Schematic of the changing bigram environment's latent probabilities (left) and sample generated sequence (right, dots: observations, lines: latent bigram probabilities). At each time step, a binary observation is randomly generated according to the relevant latent bigram probability, $p_{0|0}$ or $p_{1|1}$ depending on the previous observation. $p_{0|0}$ denotes the probability of occurrence of a 0 after a 0 and $p_{1|1}$ that of a 1 after a 1 (note that $p_{1|0}=1-p_{0|0}$ and $p_{0|1}=1-p_{1|1}$). At any time step, each of the two bigram probabilities can suddenly change to a new value uniformly drawn in [0,1], randomly with a fixed probability and independently from each other. (**b**) Example prediction sequence illustrating each network's ability or inability to change prediction according to the local context, compared to the optimal prediction (dots: observations, lines: predictions). (**c**) Prediction performance of each type of agent in the changing bigram environment. 20 new agents of each type were trained and tested as in *Figure 2b* but now in the changing bigram environment (dots: agents; bars: average). The gated recurrent network significantly outperformed every other type of agent (p < 0.001, two-tailed two independent samples t-test with Welch's correction for unequal variances). (**d**) Internalization of the latent structure as shown on an out-of-sample sequence: the two bigram probabilities are simultaneously represented in the gated recurrent network (top), and closely follow the optimal estimates (bottom). The readouts were obtained through linear regression from the recurrent activity to four estimates separately: the log odds of the mean and the log precision of the optimal posterior distribution on $p_{0|0}$ and $p_{1|1}$. In (**b**) and (**d**), the networks (out of 20) yielding median performance were selected for illustration purposes.

The online version of this article includes the following figure supplement(s) for figure 6:

**Figure supplement 1.** Performance across training and test environments.

Is the networks' representation of the latent bigram structure impenetrable or easily accessible? We tested the latter possibility by trying to linearly read out the optimal estimate of each of the latent bigram probabilities from the recurrent activity of a gated recurrent network (see Materials and methods). Arguing in favor of an explicit representation, we found that the read estimates of each of the latent bigram probabilities on left-out data were highly accurate (Pearson correlation with the optimal estimates, median and CI: 0.97 [0.97, 0.98] for each of the two bigram probabilities).

In addition to the point estimates of the latent bigram probabilities, we also tested whether a network maintained some information about the precision of each estimate. Again, we assessed the possibility to linearly read out the optimal precision of each estimate and found that the read precisions on left-out data were quite accurate (Pearson correlation with the optimal precisions, median and CI: 0.77 [0.74, 0.78] for one bigram probability and 0.76 [0.74, 0.78] for the other probability).

*Figure 6d* illustrates the striking resemblance between the estimates read from a gated recurrent network and the optimal estimates. Furthermore, it shows that the network successfully disentangles one bigram probability from the other since the read estimates can evolve independently from each other (for instance during the first 20 time steps, the value for 1|1 changes while the value for 0|0 does not, since only 1s are observed). It is particularly interesting that both bigram probabilities are simultaneously represented, given that only one of them is relevant for the moment-by-moment prediction read by the network's output unit (whose weights cannot change during the sequence).

We conclude that gated recurrent networks internalize the latent bigram structure in such a way that both bigram probabilities are available simultaneously, even though only one of the two is needed at any one time for the prediction.

## Leveraging a higher-level structure: inference about latent changes

In real life, latent structures can also exhibit different levels that are organized hierarchically (*Bill et al., 2020*; *Meyniel et al., 2015*; *Purcell and Kiani, 2016*). To study the ability to leverage such a hierarchical structure, we designed a third environment in which, in addition to bigram probabilities, we introduced a higher-level factor: the change points of the two bigram probabilities are now coupled, rather than independent as they were in the previous environment (*Figure 7a*; *Figure 1—figure supplement 1* shows the hierarchical structure). Due to this coupling, from the agent's point of view, the likelihood that a change point has occurred depends on the observations about both bigrams. Thus, optimal prediction requires the ability to make a higher-level inference: having observed that the frequency of one of the bigrams has changed, one should not only suspect that the latent probability of this bigram has changed but also transfer this suspicion of a change to the latent probability of the other bigram, even without any observations about that bigram.

Such a transfer has been reported in humans (*Heilbron and Meyniel, 2019*, Figure 5B). A typical situation is when a streak of repetitions is encountered (*Figure 7b*): if a long streak of 1 s was deemed unlikely, it should trigger the suspicion of a change point such that p(1|1) is now high, and this suspicion should be transferred to p(0|0) by partially resetting it. This reset is reflected in the change between the prediction following the 0 just before the streak and that following the 0 just after the streak (*Figure 7b*, $|p_{after}-p_{before}|$).

We tested the networks' ability for higher-level inference in the same way, by exposing them to such streaks of repetitions and measuring their change in prediction about the unobserved bigram before and after the streak. More accurately, we compared the change in prediction of the networks trained in the environment with coupled change points to that of the networks trained in the environment with independent change points, since the higher-level inference should only be made in the coupled case.

We found that gated recurrent networks trained in the coupled environment changed their prediction about the unobserved bigram significantly more than networks trained in the independent environment, and this was true across a large range of streak lengths (*Figure 7c*, top plot). The mere presence of this effect is particularly impressive given that the coupling makes very little difference in terms of raw performance (*Figure 6—figure supplement 1*, the networks trained in either the coupled or the independent environment perform very similarly when tested in either environment). All mechanisms of the gated recurrence are important to achieve this higher-level inference since the networks deprived of either gating, lateral connections, or recurrent weight training did not show any effect, no matter the streak length (*Figure 7c*, bottom three plots; for every mechanism, there was a

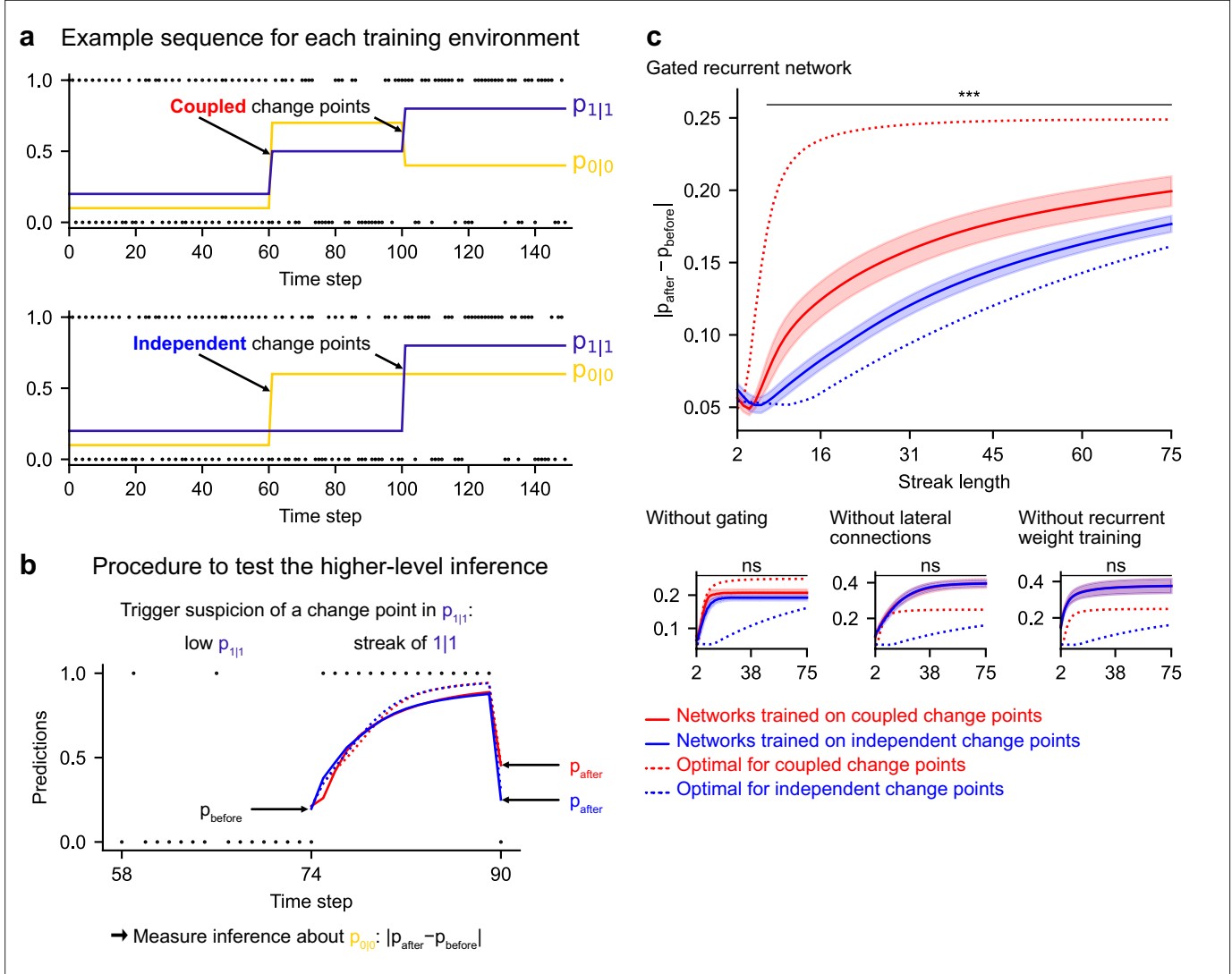

**Figure 7.** Gated recurrent but not alternative networks leverage a higher-level structure, distinguishing the case where change points are coupled vs. independent. Procedure to test the higher-level inference: (**a**) For each network architecture, 20 networks were trained on sequences where the change points of the two latent bigram probabilities are coupled and 20 other networks were trained on sequences where they are independent (the plots show an example training sequence for each case); (**b**) The networks were then tested on sequences designed to trigger the suspicion of a change point in one bigram probability and measure their inference about the other bigram probability: $|p_{after}-p_{before}|$ should be larger when the agent assumes change points to be coupled rather than independent. The plot shows an example test sequence. Red, blue, solid, and dashed lines: as in (**c**), except that only the gated recurrent network (out of 20) yielding median performance is shown for illustration purposes. (**c**) Change in prediction about the unobserved bigram probability of the networks trained on coupled change points (red) and independent change points (blue) for each network architecture, averaged over sequences. Solid lines and bands show the mean and the 95% confidence interval of the mean over networks. Dotted lines show the corresponding values of the optimal agent for the two cases. Only the gated recurrent architecture yields a significant difference between networks trained on coupled vs. independent change points (one-tailed two independent samples t-test, ***: $p < 0.001$, n.s.: $p > 0.05$).

significant interaction effect between the removal of the mechanism and the training environment on the change in prediction over networks and streak lengths, all $F_{(1,6076)} > 43.2$, all $p < 0.001$.

These results show that gated recurrent networks but not alternative networks leverage the higher level of structure where the change points of the latent probabilities are coupled.

## Gated recurrence enables simple solutions

Finally, we highlight the small number of units sufficient to perform quasi-optimally in the increasingly structured environments that we tested: the above-mentioned results were obtained with 11 recurrent units. It turns out that gated recurrent networks can reach a similar performance with even fewer units,

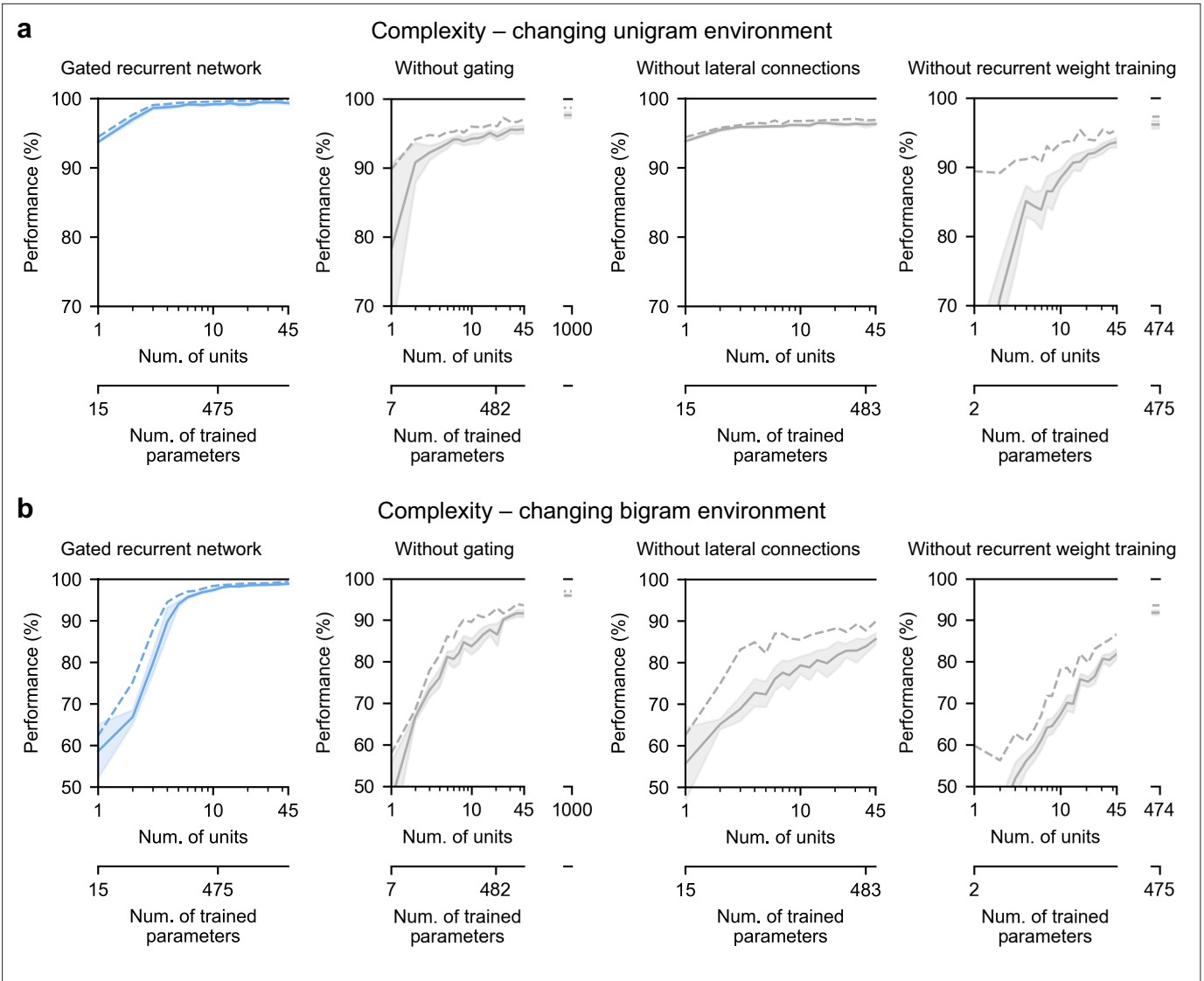

**Figure 8.** Low-complexity solutions are uniquely enabled by the combination of gating, lateral connections, and recurrent weight training. (**a** and **b**) Prediction performance of each network architecture in the changing unigram environment and the changing bigram environment, respectively, as a function of the number of recurrent units (i.e. space complexity) of the network. For each network architecture and each number of units, 20 networks were trained using hyperparameters that had been optimized prior to training, and prediction performance was measured as the % of optimal log likelihood on new test sequences. Solid lines, bands, and dashed lines show the mean, 95% confidence interval of the mean, and maximum performance, respectively. At the maximum displayed number of units, all of the alternative architectures have exceeded the complexity of the 11-unit gated recurrent network shown on the left and in previous Figures, both in terms of the number of units and the number of trained parameters (indicated on the twin x-axes), but none of them have yet reached its performance.

The online version of this article includes the following figure supplement(s) for figure 8:

**Figure supplement 1.** Training speed of the gated recurrent networks in the changing unigram and bigram environments.

especially in simpler environments (*Figure 8a and b*, left plot). For instance, in the unigram environment, gated recurrent networks reach 99% of their asymptotic performance with no more than 3 units.

By contrast, without either gating, lateral connections, or recurrent weight training, even when the networks are provided with more units to match the number of trained parameters in the 11-unit gated recurrent networks, they are unable to achieve similar performance (*Figure 8a and b*, right three plots, the twin x-axes indicate the number of units and trained parameters).

With an unlimited number of units, at least in the case without gating (i.e. a vanilla RNN, short for recurrent neural network), the networks will be able to achieve such performance since they are

universal approximators of dynamical systems (*Cybenko, 1989*; *Schäfer and Zimmermann, 2006*). However, our results indicate that this could require a very large number of units even in the simplest environment tested here (see *Figure 8a and b*, without gating at 1000 units). Indeed, the slow growth of the vanilla RNNs' performance with the number of units is well described by a power law function, of the form: $(100-p) = c(1/N)^\alpha$, where p is the % of optimal performance and N is the number of units. We fitted this law in the unigram environment using the obtained performance from 2 to 45 units and it yielded a goodness-of-fit of $R^2 = 92.4\%$ (fitting was done by linear regression on the logarithm of N and $(100-p)$). To further confirm the validity of the power law, we then extrapolated to 1,000 units and found that the predicted performance was within 0.2% of the obtained performance for networks of this size (predicted: 97.8%, obtained: 97.6%). Based on this power law, more than $10^4$ units would be needed for the vanilla RNN to reach the performance exhibited by the GRU with only 11 units.

Note that, in terms of computational complexity, the number of units is a fair measure of space complexity (i.e. the amount of memory) across the architectures we considered, since in all of them it is equal to the number of state variables (having one state variable per unit, see Materials and methods). What varies across architectures is the number of trained parameters, that is, the degrees of freedom that can be used during training to achieve different dynamics. Still, the conclusion remains the same when an alternative network exceeds the complexity of an 11-unit gated recurrent network in both its number of units and its number of trained parameters.

Therefore, it is the specific computational properties provided by the combination of the three mechanisms that afford effective low-complexity solutions.

## Discussion

We have shown that the gated recurrent architecture enables simple and effective solutions: with only 11 units, the networks perform quasi-optimally in environments fraught with randomness, changes, and different levels of latent structure. Moreover, these solutions reproduce several aspects of optimality observed in organisms, including the adaptation of their effective learning rate, the ability to represent the precision of their estimation and to use it to weight their updates, and the ability to represent and leverage the latent structure of the environment. By depriving the architecture of one of its mechanisms, we have shown that three of them are important to achieve such solutions: gating, lateral connections, and the training of recurrent weights.

### Can small neural networks behave like Bayesian agents?

A central and much-debated question in the scientific community is whether the brain can perform Bayesian inference (*Knill and Pouget, 2004*; *Bowers and Davis, 2012*; *Griffiths et al., 2012*; *Rahnev and Denison, 2018*; *Lee and Mumford, 2003*; *Rao and Ballard, 1999*; *Sanborn and Chater, 2016*; *Chater et al., 2006*; *Findling et al., 2019*; *Wyart and Koechlin, 2016*; *Soltani and Izquierdo, 2019*; *Findling et al., 2021*). From a computational viewpoint, there exists no tractable solution (even approximate) for Bayesian inference in an arbitrary environment, since it is NP-hard (*Cooper, 1990*; *Dagum and Luby, 1993*). Being a bounded agent (*Simon, 1955*; *Simon, 1972*), the brain cannot solve Bayesian inference in its most general form. The interesting question is whether the brain can perform Bayesian inference in some environments that occur in real life. More precisely, by 'perform Bayesian inference' one usually means that it performs computations that satisfy certain desirable properties of Bayesian inference, such as taking into account a certain type of uncertainty and a certain type of latent structure (*Courville et al., 2006*; *Deroy et al., 2016*; *Griffiths et al., 2012*; *Knill and Pouget, 2004*; *Ma, 2010*; *Ma and Jazayeri, 2014*; *Tauber et al., 2017*). In this study, we selected specific properties and showed that they can indeed be satisfied when using specific (not all) neural architectures.

In the changing unigram and changing bigram environments, our results provide an existence proof: there exist plausible solutions that are almost indistinguishable from Bayesian inference (i.e. the optimal solution). They exhibit qualitative properties of Bayesian inference that have been demonstrated in humans but are lacking in heuristic solutions, such as the dynamic adjustment of the effective learning rate (*Behrens et al., 2007*; *Nassar et al., 2010*; *Nassar et al., 2012*), the internal representation of latent variables and the precision of their estimates (*Boldt et al., 2019*; *Meyniel et al., 2015*), the precision-weighting of updates (*McGuire et al., 2014*; *Nassar et al., 2010*; *Nassar*

*et al., 2012*), and the ability for higher-level inference (*Bill et al., 2020*; *Heilbron and Meyniel, 2019*; *Purcell and Kiani, 2016*).

The performance we obtained with the gated recurrent architecture is consistent with the numerous other successes it produced in other cognitive neuroscience tasks (*Wang et al., 2018*; *Yang et al., 2019*; *Zhang et al., 2020*). Our detailed study reveals that it offers quasi-optimal low-complexity solutions to new and difficult challenges, including those posed by bigram and higher-level structures and latent probabilities that change unpredictably anywhere in the unit interval. We acknowledge that further generalization to additional challenges remains to be investigated, including the use of more than two categories of observations or continuous observations, and latent structures with longer range dependencies (beyond bigram probabilities).

## Minimal set of mechanisms

What are the essential mechanistic elements that enable such solutions? We show that it suffices to have recurrent units of computation equipped with three mechanisms: (1) input, self, and lateral connections which enable each unit to sum up the input with their own and other units' prior value before a non-linear transformation is applied; (2) gating, which enables multiplicative interactions between activities at the summation step; (3) the training of connection weights.

One of the advantages of such mechanisms is their generic character: they do not include any components specifically designed to perform certain probabilistic operations or estimate certain types of latent variables, as often done in neuroscience (*Echeveste et al., 2020*; *Fusi et al., 2007*; *Jazayeri and Movshon, 2006*; *Ma et al., 2006*; *Pecevski et al., 2011*; *Soltani and Wang, 2010*). In addition, they allow adaptive behavior only through recurrent activity dynamics, without involving synaptic plasticity as in other models (*Farashahi et al., 2017*; *Fusi et al., 2005*; *Iigaya, 2016*; *Schultz et al., 1997*). This distinction has implications for the timescale of adaptation: in the brain, recurrent dynamics and synaptic plasticity often involve short and long timescales, respectively. Our study supports this view: recurrent dynamics allow the networks to quickly adapt to a given change in the environment (*Figure 3*), while synaptic plasticity allows the training process to tune the speed of this adaptation to the frequency of change of the environment (*Figure 3—figure supplement 1*).

Our findings suggest that these mechanisms are particularly advantageous to enable solutions with low computational complexity. Without one of them, it seems that a very large number of units (i.e. a large amount of memory) would be needed to achieve comparable performance (*Figure 8*) (note that universal approximation bounds in vanilla RNNs can be very large in terms of number of units [*Barron, 1993*; *Cybenko, 1989*; *Schäfer and Zimmermann, 2006*]). These mechanisms thus seem to be key computational building blocks to build simple and effective solutions. This efficiency can be formalized as the minimum number of units sufficient for near-optimal performance (as in *Orhan and Ma, 2017* who made a similar argument), and it is important for the brain since the brain has limited computational resources (often quantified by the Shannon capacity, i.e. the number of bits that can be transmitted per unit of time, which here amounts to the number of units) (*Bhui et al., 2021*; *Lieder and Griffiths, 2019*). Moreover, simplicity promotes our understanding, and it is with the same goal of understanding that others have used model reduction in large networks (*Dubreuil et al., 2020*; *Jazayeri and Ostojic, 2021*; *Schaeffer et al., 2020*).

Since we cannot exhaustively test all possible parameter values, it might be possible that better solutions exist that were not discovered during training. However, to maximize the chances that the best possible performance is achieved after training, we conducted an extensive hyperparameter optimization, repeated for each environment, architecture, and several number of units, until there is no more improvement according to the Bayesian optimization (see Materials and methods).

## Biological implementations of the mechanisms

What biological elements could implement the mechanisms of the gated recurrence? Recurrent connections are ubiquitous in the brain (*Douglas and Martin, 2007*; *Hunt and Hayden, 2017*); the lesser-known aspect is that of gating. In the next paragraph, we speculate on the possible biological implementations of gating, broadly defined as a mechanism that modulates the effective weight of a connection as a function of the network state (and not limited to the very specific form of gating of the GRU).

In neuroscience, many forms of gating have been observed, and they can generally be grouped into three categories according to the neural process that supports them: neural circuits, neural oscillations, and neuromodulation. In neural circuits, a specific pathway can be gated through inhibition/disinhibition by inhibitory (GABAergic) neurons. This has been observed in microscopic circuits, e.g. in pyramidal neurons a dendritic pathway can be gated by interneurons (*Costa et al., 2017*; *Yang et al., 2016*), or macroscopic circuits, for example in basal ganglia-thalamo-cortical circuits a cortico-cortical pathway can be gated by the basal ganglia and the mediodorsal nucleus of thalamus (*O'Reilly, 2006*; *O'Reilly and Frank, 2006*; *Rikhye et al., 2018*; *Wang and Halassa, 2021*; *Yamakawa, 2020*). In addition to inhibition/disinhibition, an effective gating can also be achieved by a large population of interacting neurons taking advantage of their nonlinearity (*Beiran et al., 2021*; *Dubreuil et al., 2020*). Regarding neural oscillations, experiments have shown that activity in certain frequency bands (typically, alpha and beta) can gate behavioral and neuronal responses to the same stimulus (*Baumgarten et al., 2016*; *Busch et al., 2009*; *Hipp et al., 2011*; *Iemi et al., 2019*; *Klimesch, 1999*; *Mathewson et al., 2009*). One of the most influential accounts is known as 'pulsed inhibition' (*Hahn et al., 2019*; *Jensen and Mazaheri, 2010*; *Klimesch et al., 2007*): a low-frequency signal periodically inhibits a high-frequency signal, effectively silencing the high-frequency signal when the low-frequency signal exceeds a certain threshold. Finally, the binding of certain neuromodulators to the certain receptors of a synapse changes the gain of its input-output transfer function, thus changing its effective weight. This has been demonstrated in neurophysiological studies implicating noradrenaline (*Aston-Jones and Cohen, 2005*; *Salgado et al., 2016*; *Servan-Schreiber et al., 1990*), dopamine (*Moyer et al., 2007*; *Servan-Schreiber et al., 1990*; *Stalter et al., 2020*; *Thurley et al., 2008*), and acetylcholine (*Gil et al., 1997*; *Herrero et al., 2008*) (see review in *Thiele and Bellgrove, 2018*).

We claim that gated recurrence provides plausible solutions for the brain because its mechanisms can all be biologically implemented and lead to efficient solutions. However, given their multiple biological realizability, the mapping between artificial units and biological neurons is not straightforward: one unit may map to a large population of neurons (e.g. a brain area), or even to a microscopic, subneuronal component (e.g. the dendritic level).

## Training: Its role and possible biological counterpart

Regarding the training, our results highlight that it is important to adjust the recurrent weights and thus the network dynamics to the environment (and not fix them as in reservoir computing [*Tanaka et al., 2019*]), but we make no claims about the biological process that leads to such adjustment in brains. It could occur during development (*Sherman et al., 2020*), the life span (*Lillicrap et al., 2020*), or the evolution process (*Zador, 2019*) (these possibilities are not mutually exclusive). Although our training procedure may not be accurate for biology as a whole, two aspects of it may be informative for future research. First, it relies only on the observation sequence (no supervision or reinforcement), leveraging prediction error signals, which have been found in the brain in many studies (*den Ouden et al., 2012*; *Eshel et al., 2013*; *Maheu et al., 2019*). Importantly, in predictive coding (*Rao and Ballard, 1999*), the computation of prediction errors is part of the prediction process; here we are suggesting that it may also be part of the training process (as argued in *O'Reilly et al., 2021*). Second, relatively few iterations of training suffice (*Figure 8—figure supplement 1*, in the order of 10–100; for comparison, *Wang et al., 2018* reported training for 40,000 episodes in an environment similar to ours).

## Suboptimalities in human behavior

In this study we have focused on some aspects of optimality that humans exhibit in the three environments we explored, but several aspects of their behavior are also suboptimal. In the laboratory, their behavior is often at best qualitatively Bayesian but quantitatively suboptimal. For example, although they adjust their effective learning rate to changes, the base value of their learning rate and their dynamic adjustments may depart from the optimal values (*Nassar et al., 2010*; *Nassar et al., 2012*; *Prat-Carrabin et al., 2021*). They may also not update their prediction on every trial, unlike the optimal solution (*Gallistel et al., 2014*; *Khaw et al., 2017*). Finally, there is substantial interindividual variability which does not exist in the optimal solution (*Khaw et al., 2021*; *Nassar et al., 2010*; *Nassar et al., 2012*; *Prat-Carrabin et al., 2021*). In the future, these suboptimalities could be explored using our networks by making them suboptimal in three ways (among others): by stopping

training before quasi-optimal performance is reached (*Caucheteux and King, 2021*; *Orhan and Ma, 2017*), by constraining the size of the network or its weights (with hard constraints or with regularization penalties) (*Mastrogiuseppe and Ostojic, 2017*; *Sussillo et al., 2015*), or by altering the network in a certain way, such as pruning some of the units or some of the connections (*Blalock et al., 2020*; *Chechik et al., 1999*; *LeCun et al., 1990*; *Srivastava et al., 2014*), or introducing random noise into the activity (*Findling et al., 2021*; *Findling and Wyart, 2020*; *Legenstein and Maass, 2014*). In this way, one could perhaps reproduce the quantitative deviations from optimality while preserving the qualitative aspects of optimality observed in the laboratory.

## Implications for experimentalists

If already trained gated recurrent networks exist in the brain, then one can be used in a new but similar enough environment without further training. This is an interesting possibility because, in laboratory experiments mirroring our study, humans perform reasonably well with almost no training but explicit task instructions given in natural language, along with a baggage of prior experience (*Gallistel et al., 2014*; *Heilbron and Meyniel, 2019*; *Khaw et al., 2021*; *Meyniel et al., 2015*; *Peterson and Beach, 1967*). In favor of the possibility to reuse an existing solution, we found that a gated recurrent network can still perform well in conditions different from those it was trained in: across probabilities of change points (*Figure 3—figure supplement 1*) and latent structures (*Figure 6—figure supplement 1*, from bigram to unigram).

In this study, we adopted a self-supervised training paradigm to see if the networks could in principle discover the latent structure from the sequences of observations alone. However, in laboratory experiments, humans often do not have to discover the structure since they are explicitly told what structure they will face and the experiment starts only after ensuring that they have understood it, which makes the comparison to our networks impossible in this setting in terms of training (see similar argument in *Orhan and Ma, 2017*). In the future, it could be interesting to study the ability of gated recurrent networks to switch from one structure to another after having been informed of the current structure as humans do in these experiments. One possible way would be to give a label that indicates the current structure as additional input to our networks, as in *Yang et al., 2019*.

One of our findings may be particularly interesting to experimentalists: in a gated recurrent network, the representations of latent probabilities and the precision of these probability estimates (sometimes referred to as confidence [*Boldt et al., 2019*; *Meyniel et al., 2015*], estimation uncertainty [*McGuire et al., 2014*; *Payzan-LeNestour et al., 2013*], or epistemic uncertainty [*Amini et al., 2020*; *Friston et al., 2015*; *Pezzulo et al., 2015*]) are linearly readable from recurrent activity, the form of decoding most frequently used in neuroscience (*Haxby et al., 2014*; *Kriegeskorte and Diedrichsen, 2019*). These representations arise spontaneously, and their emergence seems to come from the computational properties of gated recurrence together with the need to perform well in a stochastic and changing environment. This yields an empirical prediction: if such networks can be found in the brain, then latent probability estimates and their precision should also be decodable in brain signals, as already found in some studies (*Bach et al., 2011*; *McGuire et al., 2014*; *Meyniel, 2020*; *Meyniel and Dehaene, 2017*; *Payzan-LeNestour et al., 2013*; *Tomov et al., 2020*).

## Materials and methods
### Sequence prediction problem

The sequence prediction problem to be solved is the following. At each time step, an agent receives as input a binary-valued 'observation', $x_t \in \{0, 1\}$, and gives as output a real-valued 'prediction', $p_t \in [0, 1]$ which is an estimate of the probability that the value of the next observation is equal to 1, $p(x_{t+1} = 1)$. Coding the prediction in terms of the observation being 1 rather than 0 is inconsequential since one can be deduced from the other: $p(x_{t+1} = 1) = 1 - p(x_{t+1} = 0)$. The agent's objective is to make predictions that maximize the (log) likelihood of observations in the sequence, which technically corresponds to the negative binary cross-entropy cost function:

$$L(p; x) = \sum_{t=0}^{T-1} \log[x_{t+1}p_t + (1 - x_{t+1})(1 - p_t)] \qquad (1)$$

## Network architectures

All network architectures consist of a binary input unit, which codes for the current observation, one recurrent layer (sometimes called hidden layer) with a number N of recurrent units, and an output unit, which represents the network's prediction. Unless otherwise stated, N = 11. At every time step, the recurrent unit $i$ receives as input the value of the observation, $x_t$, and the previous activation values of the recurrent units $j$ that connect to $i$, $h_{j,t-1}$. It produces as output a new activation value, $h_{i,t}$, which is a real number. The output unit receives as input the activations of all of the recurrent units, and produces as output the prediction $p_t$.

The parameterized function of the output unit is the same for all network architectures:

$$p_t = \sigma \left( \sum_{i=1}^{N} w_{hp,i} h_{i,t} + b_{hp} \right)$$

where $\sigma$ is the logistic sigmoid, $w_{hp,i}$ is the weight parameter of the connection from the $i$-th recurrent unit to the output unit, and $b_{hp}$ is the bias parameter of the output unit.

The updating of $h_i$ takes a different form depending on whether gating or lateral connections are included, as described below.

### Gated recurrent network

A gated recurrent network includes both gating and lateral connections. This enables multiplicative interactions between the input and recurrent activity as well as the activities of different recurrent units during the updating of $h_i$. The variant of gating used here is GRU (*Cho et al., 2014*; *Chung et al., 2014*). For convenience of exposition, we introduce, for each recurrent unit $i$, two intermediate variables in the calculation of the update: the reset gate $r_i$ and the update gate $z_i$, both of which have their own set of weights and bias. The update gate corresponds to the extent to which a unit can change its values from one time step to the next, and the reset gate corresponds to the balance between recurrent activity and input activity in case of update. Note that $r_i$ and $z_i$ do not count as state variables since the system would be equivalently characterized without them by injecting their expression into the update equation of $h_i$ below. The update is calculated as follows:

$$r_{i,t+1} = \sigma \left( w_{xr,i} x_{t+1} + b_{xr,i} + w_{hr,ii} h_{i,t} + \sum_{j \neq i} w_{hr,ji} h_{j,t} + b_{hr,i} \right)$$
$$z_{i,t+1} = \sigma \left( w_{xz,i} x_{t+1} + b_{xz,i} + w_{hz,ii} h_{i,t} + \sum_{j \neq i} w_{hz,ji} h_{j,t} + b_{hz,i} \right)$$
$$h_{i,t+1} = z_{i,t+1} h_{i,t}$$
$$+ (1 - z_{i,t+1}) \tanh \left[ w_{xh,i} x_{t+1} + b_{xh,i} + r_{i,t+1} (w_{hh,ii} h_{i,t} + \sum_{j \neq i} w_{hh,ji} h_{j,t}) + b_{hh,i} \right]$$
$$h_{i,t=-1} = 0$$

where $(w_{xr,i}, b_{xr,i}, w_{hr,ji}, b_{hr,i})$, $(w_{xz,i}, b_{xz,i}, w_{hz,ji}, b_{hz,i})$, $(w_{xh,i}, b_{xh,i}, w_{hh,ji}, b_{hh,i})$ are the connection weights and biases from the input unit and the recurrent units to unit $i$ corresponding to the reset gate, the update gate, and the ungated new activity, respectively.

Another variant of gating is the LSTM (*Hochreiter and Schmidhuber, 1997*). It incorporates similar gating mechanisms as that of the GRU and can achieve the same performance in our task. We chose the GRU because it is simpler than the LSTM and it turned out sufficient.

### Without gating

Removing the gating mechanism from the gated recurrent network is equivalent to setting the above variables $r_i$ equal to 1 and $z_i$ equal to 0. This simplifies the calculation of the activations to a single equation, which boils down to a weighted sum of the input and the recurrent units' activity before applying a non-linearity, as follows:

$$h_{i,t+1} = \tanh \left[ w_{xh,i} x_{t+1} + b_{xh,i} + w_{hh,ii} h_{i,t} + \sum_{j \neq i} w_{hh,ji} h_{j,t} + b_{hh,i} \right]$$

Another possibility (not considered here) would be to set the value of $z_i$ to a constant other than 1 and treat this value (which amounts to a time constant) as a hyperparameter.

## Without lateral connections

Removing lateral connections from the gated recurrent network is equivalent to setting the weights $w_{hr,ji}$, $w_{hz,ji}$, and $w_{hh,ji}$ to 0 for all $j \neq i$. This abolishes the possibility of interaction between recurrent units, which simplifies the calculation of the activations as follows:

$$
\begin{aligned}
r_{i,t+1} &= \sigma(w_{xr,i}x_{t+1} + b_{xr,i} + w_{hr,ii}h_{i,t} + b_{hr,i}) \\
z_{i,t+1} &= \sigma(w_{xz,i}x_{t+1} + b_{xz,i} + w_{hz,ii}h_{i,t} + b_{hz,i}) \\
h_{i,t+1} &= z_{i,t+1}h_{i,t} + (1 - z_{i,t+1})\tanh[w_{xh,i}x_{t+1} + r_{i,t+1}w_{hh,ii}h_{i,t} + b_{hh,i}]
\end{aligned}
$$

Note that this architecture still contains gating. We could have tested a simpler architecture without lateral connection and without gating; however, our point is to demonstrate the specific importance of lateral connections to solve the problem we are interested in with few units, and the result is all the more convincing if the network lacking lateral connections has gating (without gating, it would fail even more dramatically).

## Without recurrent weight training

The networks referred to as 'without recurrent weight training' have the same architecture as the gated recurrent networks and differ from them only in the way they are trained. While in the other networks, all of the weights and bias parameters are trained, for those networks, only the weights and bias of the output unit, $w_{hp,i}w_{hp,i}$ and $b_{hp}$, are trained; other weights and biases are fixed to the value drawn at initialization.

# Environments

An environment is characterized by its data generating process, that is, the stochastic process used to generate a sequence of observations in that environment. Each of the generative processes is described by a graphical model in *Figure 1—figure supplement 1* and further detailed below.

## Changing unigram environment

In the changing unigram environment, at each time step, one observation is drawn from a Bernoulli distribution whose probability parameter is the latent variable $p_t^{env}$. The evolution of this latent variable is described by the following stochastic process.

- Initially, $p_{t=0}^{env}$ is drawn from a uniform distribution on [0,1].
- At the next time step, with probability $p_c$, $p_{t+1}^{env}$ is drawn anew from a uniform distribution on [0,1] (this event is called a 'change point'), otherwise, $p_{t+1}^{env}$ remains equal to $p_t^{env}$. The change point probability $p_c$ is fixed in a given environment.

## Changing bigram environments

In the changing bigram environments, at each time step, one observation is drawn from a Bernoulli distribution whose probability parameter is either equal to the latent variable $p_{1|1,t}^{env}$, if the previous observation was equal to 1, or to the latent variable $(1 - p_{0|0,t}^{env})$ otherwise (at t = 0, the previous observation is considered to be equal to 0). The evolution of those latent variables is described by a stochastic process which differs depending on whether the change points are independent or coupled.

- In both cases, initially, $p_{0|0,t=0}^{env}$ and $p_{1|1,t=0}^{env}$ are both drawn independently from a uniform distribution on [0,1].
- In the case of *independent change points*, at the next time step, with probability $p_c$, $p_{0|0,t+1}^{env}$ is drawn anew from a uniform distribution on [0,1], otherwise, $p_{0|0,t+1}^{env}$ remains equal to $p_{0|0,t}^{env}$. Similarly, $p_{1|1,t+1}^{env}$ is either drawn anew with probability $p_c$ or remains equal to $p_{1|1,t}^{env}p_{1|1,t}^{env}$ otherwise, and critically, the occurrence of a change point in $p_{1|1}^{env}$ is independent from the occurrence of a change point in $p_{0|0}^{env}$.
- In the case of *coupled change points*, at the next time step, with probability $p_c$, $p_{0|0,t+1}^{env}$ and $p_{1|1,t+1}^{env}$ are both drawn anew and independently from a uniform distribution on [0,1], otherwise, both remain equal to $p_{0|0,t}^{env}$ and $p_{1|1,t}^{env}$ respectively.

The changing bigram environment with independent change points and that with coupled change points constitute two distinct environments. When the type of change points is not explicitly mentioned,

the default case is independent change points. For conciseness, we sometimes refer to the changing unigram and changing bigram environments simply as 'unigram' and 'bigram' environments.

In all environments, unless otherwise stated, the length of a sequence is $T = 380$ observations, and the change point probability is $p_c = \frac{1}{75}$, as in previous experiments done with human participants (*Heilbron and Meyniel, 2019*; *Meyniel et al., 2015*).

## Optimal solution

For a given environment among the three possibilities defined above, the optimal solution to the prediction problem can be determined as detailed in *Heilbron and Meyniel, 2019*. This solution consists in inverting the data-generating process of the environment using Bayesian inference, that is, computing the posterior probability distribution over the values of the latent variables given the history of observation values, and then marginalizing over that distribution to compute the prediction (which is the probability of the next observation given the history of observations). This can be done using a hidden Markov model formulation of the data-generating process where the hidden state includes the values of the latent variables as well as the previous observation in the bigram case, and using the forward algorithm to compute the posterior distribution over the hidden state. Because it would be impossible to compute the probabilities for the infinitely many possible values of the latent variables in the continuous interval [0,1], we discretized the interval into 20 equal-width bins for each of the latent variables. For a more exhaustive treatment, see *Heilbron and Meyniel, 2019* and the online code (https://github.com/florentmeyniel/TransitionProbModel).

## Heuristic solutions

The four heuristic solutions used here can be classified into 2 × 2 groups depending on:

- which kind of variables are estimated: a *unigram probability* or two *bigram probabilities*.
- which heuristic rule is used in the calculation of the estimates: the *delta-rule* or the *leaky rule*.

The equations used to calculate the estimates are provided below.

*Unigram, delta-rule*:
$$\hat{p}_{t+1} = \hat{p}_t + \alpha(x_{t+1} - \hat{p}_t)$$
$$\hat{p}_{t=-1} = 0.5$$
*Unigram, leaky rule*:
$$n_{0,t+1} = \alpha n_{0,t} + (1 - x_{t+1})$$
$$n_{1,t+1} = \alpha n_{1,t} + x_{t+1}$$
$$n_{0,t=-1} = n_{1,t=-1} = 0$$
$$\hat{p}_t = \frac{n_{1,t}+1}{n_{1,t}+n_{0,t}+2}$$
*Bigrams, delta-rule*:
$$\hat{p}_{0|0,t+1} = \hat{p}_{0|0,t} + \alpha(1 - x_t)(1 - x_{t+1} - \hat{p}_{0|0,t})$$
$$\hat{p}_{1|1,t+1} = \hat{p}_{1|1,t} + \alpha x_t(x_{t+1} - \hat{p}_{1|1,t})$$
$$\hat{p}_{0|0,t=-1} = \hat{p}_{1|1,t=-1} = 0.5$$
*Bigrams, leaky rule*:
$$n_{0|0,t+1} = \alpha n_{0|0,t} + (1 - x_t)(1 - x_{t+1})$$
$$n_{1|0,t+1} = \alpha n_{1|0,t} + (1 - x_t)x_{t+1}$$
$$n_{0|1,t+1} = \alpha n_{0|1,t} + x_t(1 - x_{t+1})$$
$$n_{1|1,t+1} = \alpha n_{1|1,t} + x_t x_{t+1}$$
$$n_{0|0,t=-1} = n_{1|0,t=-1} = n_{0|1,t=-1} = n_{1|1,t=-1} = 0$$
$$\hat{p}_{0|0,t} = \frac{n_{0|0,t}+1}{n_{0|0,t}+n_{1|0,t}+2}$$
$$\hat{p}_{1|1,t} = \frac{n_{1|1,t}+2}{n_{1|1,t}+n_{0|1,t}+2}$$

The delta-rule corresponds to the update rule of the Rescorla-Wagner model (*Rescorla and Wagner, 1972*). The leaky rule corresponds to the mean of an approximate posterior which is a Beta distribution whose parameters depend on the leaky counts of observations: $n_1 + 1$ and $n_0 + 1$ (see *Meyniel et al., 2016* for more details).

The output prediction value is equal to $\hat{p}_t$ in the unigram case, and in the bigram case, to $\hat{p}_{1|1,t}$ if $x_t = 1$ and $(1 - \hat{p}_{0|0,t})$ otherwise. The parameter $\alpha$ is a free parameter which is trained (using the same training data as the networks) and thus adjusted to the training environment.

## Training

For a given environment and a given type of agent among the network types and heuristic types, all the reported results are based on 20 agents, each sharing the same set of hyperparameters and initialized with a different random seed. During training, the parameters of a given agent were adjusted to minimize the binary cross-entropy cost function (see *Equation 1*). During one iteration of training, the gradients of the cost function with respect to the parameters are computed on a subset of the training data (called a minibatch) using backpropagation through time and are used to update the parameters according to the selected training algorithm. The training algorithm was Adam (*Kingma and Ba, 2015*) for the network types and stochastic gradient descent for the heuristic types.

For the unigram environment, the analyses reported in *Figures 2–5* were conducted after training on a common training dataset of 160 minibatches of 20 sequences. For each of the two bigram environments, the analyses reported in *Figures 6–7* were conducted after training on a common training dataset (one per environment) of 400 minibatches of 20 sequences. These sizes were sufficient for the validation performance to converge before the end of training for all types of agents.

**Table 1.** Selected hyperparameter values after optimization.
(*: fixed value.)

| Environment | Network architecture | N | $\eta_0$ | $\sigma_{0,x\cdot}$ | $\sigma_{0,h\cdot}$ | $\mu_{0,h\cdot,ii}$ |
|---|---|---|---|---|---|---|
| unigram | gated recurrent network | 3 | 8.00E-02 | 0.02 | 0.02 | 0* |
| unigram | gated recurrent network | 11 | 6.60E-02 | 0.43 | 0.21 | 0* |
| unigram | gated recurrent network | 45 | 4.20E-02 | 1 | 0.02 | 0* |
| unigram | without gating | 3 | 2.50E-02 | 1 | 0.07 | 0* |
| unigram | without gating | 11 | 1.70E-02 | 1 | 0.07 | 0* |
| unigram | without gating | 45 | 7.60E-03 | 1 | 0.08 | 0* |
| unigram | without gating | 1,000 | 1.34E-04 | 1 | 0.04 | 0* |
| unigram | without lateral connections | 3 | 5.30E-02 | 0.02 | 0.02 | 1 |
| unigram | without lateral connections | 11 | 2.70E-02 | 1 | 0.02 | 1 |
| unigram | without lateral connections | 45 | 1.30E-02 | 1 | 1 | 1 |
| unigram | without recurrent weight training | 3 | 1.00E-01 | 1.07 | 0.55 | 0* |
| unigram | without recurrent weight training | 11 | 1.00E-01 | 2 | 0.41 | 0* |
| unigram | without recurrent weight training | 45 | 1.00E-01 | 2 | 0.26 | 0* |
| unigram | without recurrent weight training | 474 | 9.60E-03 | 1 | 0.1 | 0* |
| bigram | gated recurrent network | 3 | 6.30E-02 | 0.02 | 1 | 0* |
| bigram | gated recurrent network | 11 | 4.40E-02 | 1 | 0.02 | 0* |
| bigram | gated recurrent network | 45 | 1.60E-02 | 1 | 0.02 | 0* |
| bigram | without gating | 3 | 5.50E-02 | 0.02 | 0.13 | 0* |
| bigram | without gating | 11 | 3.20E-02 | 1 | 0.05 | 0* |
| bigram | without gating | 45 | 8.90E-03 | 1 | 0.06 | 0* |
| bigram | without gating | 1,000 | 5.97E-05 | 1 | 0.03 | 0* |
| bigram | without lateral connections | 3 | 4.30E-02 | 1 | 0.02 | 0 |
| bigram | without lateral connections | 11 | 4.30E-02 | 1 | 1 | 0 |
| bigram | without lateral connections | 45 | 2.80E-02 | 1 | 1 | 0 |
| bigram | without recurrent weight training | 3 | 6.60E-02 | 0.73 | 0.55 | 0* |
| bigram | without recurrent weight training | 11 | 1.00E-01 | 2 | 0.45 | 0* |

## Parameters initialization

For all of the networks, the bias parameters are randomly initialized from a uniform distribution on $[-1/\sqrt{N}, +1/\sqrt{N}]$ and the weights $w_{hp,i}$ are randomly initialized from a normal distribution with standard deviation $1/\sqrt{N}$ and mean 0. For all the networks, the weights $w_{xr,i}$, $w_{xz,i}$, $w_{xh,i}$ are randomly initialized from a normal distribution with standard deviation $\sigma_{0,x}$. and mean 0, and the weights $w_{hr,ji}, w_{hz,ji}, w_{hh,ji}$ are randomly initialized from a normal distribution with standard deviation $\sigma_{0,h}$. and mean 0 for all $j \neq i$ and $\mu_{0,h,ii}$ for $j = i$. $\sigma_{0,x}$, $\sigma_{0,h}$, $\mu_{0,h\cdot,ii}$ are hyperparameters that were optimized for a given environment, type of network, and number of units as detailed in the hyperparameter optimization section (the values resulting from this optimization are listed in *Table 1*).

For the initialization of the parameter $\alpha$ in the heuristic solutions, a random value r is drawn from a log-uniform distribution on the interval $[10^{-2.5}, 10^{-0.5}]$, and the initial value of $\alpha$ is set to r in the delta-rule case or exp(-r) in the leaky rule case.

## Hyperparameter optimization

Each type of agent had a specific set of hyperparameters to be optimized. For all network types, it included the initial learning rate of Adam $\eta_0$ and the initialization hyperparameters $\sigma_{0,x}$, $\sigma_{0,h}$. For the networks without lateral connections specifically, it also included $\mu_{0,h,ii}$ (for those networks, setting it close to one can help avoid the vanishing gradient problem during training *Bengio et al., 1994*; *Sutskever et al., 2013*) for the other networks, this was set to 0. For the heuristic types, it included only the learning rate of the stochastic gradient descent. A unique set of hyperparameter values was determined for each type of agent, each environment, and, for the network types, each number of units, through the optimization described next.

We used Bayesian optimization (*Agnihotri and Batra, 2020*) with Gaussian processes and the upper confidence bound acquisition function to identify the best hyperparameters for each network architecture, environment, and number of units. During the optimization, combinations of hyperparameter values were iteratively sampled, each evaluated over 10 trials with different random seeds, for a total of 60 iterations (hence, 600 trials) for a given architecture, environment, and number of units. In each trial, one network was created, trained, and its cross-entropy was measured on independent test data. The training and test datasets used for the hyperparameter optimization procedure were not used in any other analyses. The training datasets contained respectively 160 and 400 minibatches of 20 sequences for the unigram and the bigram environment; the test datasets contained 200 sequences for each environment. We selected the combination of hyperparameter values corresponding to the iteration that led to the lowest mean test cross-entropy over the 10 trials. The selected values are listed in *Table 1*.

For the heuristic types, we used random search from a log uniform distribution in the $[10^{-6}, 10^{-1}]$ range over 80 trials to determine the optimal learning rate of the stochastic gradient descent. This led to selecting the value $3.10^{-3}$ for all heuristic types and all three environments.

## Performance analyses

All agents were tested in the environment they were trained in (except for *Figure 6—figure supplement 1* which tests cross-environment performance). We used a single test dataset per environment of 1000 sequences independent of the training dataset. The log likelihood L of a given agent was measured from its predictions according to *Equation 1*. The optimal log likelihood $L_{optimal}$ was measured from the predictions of the optimal solution for the given environment. The chance log likelihood $L_{chance}$ was measured using a constant prediction of 0.5. To facilitate the interpretation of the results, the prediction performance of the agent was expressed as the % of optimal log likelihood, defined as:

$$\frac{L - L_{chance}}{L_{optimal} - L_{chance}} \times 100$$

To test the statistical significance of a comparison of performance between two types of agents, we used a two-tailed two independent samples t-test with Welch's correction for unequal variances.

## Analysis of the effective learning rate

The instantaneous effective learning rate of an agent that updates its prediction from $p_t$ to $p_{t+1}$ upon receiving observation $x_{t+1}$ is calculated as:

$$\begin{aligned} \alpha_{t+1} &= \frac{p_{t+1}-p_t}{x_{t+1}-p_t} \\ \alpha_{t=0} &= \frac{p_0-0.5}{x_0-0.5} \end{aligned} \qquad (2)$$

We call it 'effective learning rate' because, had the agent been using a delta-rule algorithm, it would be equivalent to the learning rate of the delta-rule (as can be seen by rearranging the above formula into an update equation), and because it can be measured even if the agent uses another algorithm.

## Readout analyses

The readout of a given quantity from the recurrent units of a network consists of a weighted sum of the activation values of each unit. To determine the weights of the readout for a given network, we ran a multiple linear regression using, as input variables, the activation of each recurrent unit at a given time step $h_{i,t}$, and as target variable, the desired quantity calculated at the same time step. The regression was run on a training dataset of 900 sequences of 380 observations each (hence, 342,000 samples).

In the unigram environment, the precision readout was obtained using as desired quantity the log precision of the posterior distribution over the unigram variable calculated by the optimal solution as previously described, that is, $\psi_t = -\log \sigma_t$, where $\sigma_t$ is the standard deviation of the posterior distribution over $p_{t+1}^{env}$:

$$\sigma_t = \mathbb{SD}[p_{t+1}^{env}|x_0, ..., x_t] \qquad (3)$$

In the bigram environment, the readout of the estimate of a given bigram variable was obtained using as desired quantity the log odds of the mean of the posterior distribution over that bigram variable calculated by the optimal solution, and the readout of the precision of that estimate was obtained using the log precision of that same posterior under the above definition of precision.

In *Figure 4a*, to measure the accuracy of the readout from a given network, we calculated the Pearson correlation between the quantity read from the network and the optimal quantity on a test dataset of 100 sequences (hence, 38,000 samples), independent from any training dataset. To measure the Pearson correlation between the read precision and the subsequent effective learning rate, we used 300 out-of-sample sequences (hence, 114,000 samples). To measure the mutual information between the read precision and the prediction of the network, we also used 300 out-of-sample sequences (114,000 samples).

In *Figure 6d*, the log odds and log precision were transformed back into mean and standard deviation for visualization purposes.

## Dynamics of network activity in the prediction-precision subspace

In *Figure 4b*, the network activity (i.e. the population activity of the recurrent units in the network) was projected onto the two-dimensional subspace spanned by the prediction vector and the precision vector. The prediction vector is the vector of the weights from the recurrent units to the output unit of the network, $w_{hp}$. The precision vector is the vector of the weights of the precision readout described above, $w_{h\psi}$. For the visualization, we orthogonalized the precision vector against the prediction vector using the Gram-Schmidt process (i.e. by subtracting from the precision vector its projection onto the prediction vector), and used the orthogonalized precision vector to define the y-axis shown in *Figure 4b*.

## Perturbation experiment to test precision-weighting

The perturbation experiment reported in *Figure 5* is designed to test the causal role of the precision read from a given network on its weighting of the next observation, measured through its effective learning rate. We performed this perturbation experiment on each of the 20 networks that were trained within each of the four architectures we considered. The causal instrument is a perturbation vector q that is added to the network's recurrent unit activations. The perturbation vector was randomly generated subject to the following constraints:

- $q \cdot w_{h\psi} = \delta\psi$ is the desired change in precision (we used five levels) that is read from the units' activities; it is computed by projecting the perturbation onto the weight vector of the precision readout ($w_{h\psi}$, $\cdot$ is the dot product);

- the perturbation $q$ induces no change in the prediction of the network: $q \cdot w_{hp} = 0$, where $w_{hp}$ is the weight vector of the output unit of the network;
- the perturbation has a constant intensity c across simulations, which we formalize as the norm of the perturbation: $\|q\| = c$.

We describe below the algorithm that we used to generate random perturbations $q$ that satisfy these constraints. The idea is to decompose $q$ into two components: both components leave the prediction unaffected, the first ($q_\psi$) is used to induce a controlled change in precision, the second ($q_r$) does not change the precision but is added to ensure a constant intensity of the perturbation across simulations.

1. To ensure no change in precision, we compute Q, the subspace of the activation space spanned by all vectors q that are orthogonal to the prediction weight vector $w_{hp}$, as the null space of $w_{hp}$ (i.e. the orthogonal complement of the subspace spanned by $w_{hp}$, dimension N-1).
2. We compute $q_\psi$, the vector component of Q that affects precision, as the orthogonal projection of $w_{h\psi}$ onto Q ($q_\psi$ is thus collinear to the orthogonalized precision axis shown in *Figure 4b* and described above).
3. We compute $\beta_\psi$, the coefficient to assign to $q_\psi$ in the perturbation vector to produce the desired change in precision $\delta\psi$, as $\beta_\psi = \frac{\delta\psi}{\|q_\psi \cdot w_{h\psi}\|}$.
4. We compute R, the subspace spanned by all vector components of Q that do not affect precision, as the null space of $q_\psi$ (dimension N-2). A perturbation vector in R therefore leaves both the prediction and the precision unchanged.
5. We draw a random unit vector $q_r$ within R (by drawing from all N-2 components).
6. We compute $\beta_r$, the coefficient to assign to $q_r$ in the perturbation vector so as to ensure that the final perturbation's norm equals c, as $\beta_r = \sqrt{c^2 - \beta_\psi^2 \|q_\psi\|^2}$.
7. We combine $q_\psi$ and $q_r$ into the final perturbation vector as $q = \beta_\psi q_\psi + \beta_r q_r$.

The experiment was run on a set of 1000 sample time points randomly drawn from 300 sequences. First, the unperturbed learning rate was measured by running the network on all of the sequences. Second, for each sample time point, the network was run unperturbed up until that point, a perturbation vector was randomly generated for the desired change of precision and applied to the network at that point, then the perturbed network was run on the next time point and its perturbed learning rate was measured. This was repeated for each level of change in precision. Finally, for a given change in precision, the change in learning rate was calculated as the difference between the perturbed and the unperturbed learning rate.

For statistical analysis, we ran a one-tailed paired t-test to test whether the population's mean change in learning rate was higher at one level of precision change than at the next level of precision change. This was done for each of the four consecutive pairs of levels of change in precision.

## Test of higher-level inference about changes

For a given network architecture, higher-level inference about changes was assessed by comparing the population of 20 networks trained in the environment with coupled change points to the population of 20 networks trained in the environment with independent change points.

In *Figure 7c*, the change in unobserved bigram prediction for a given streak length *m* was computed as follows. First, prior sequences were generated and each network was run on each of the sequences. We generated initial sequences of 74 observations each with a probability of 0.2 for the 'observed' bigram (which will render its repetition surprising) and a probability *p* for the 'unobserved' bigram equal to 0.2 or 0.8 (such probabilities, symmetric and substantially different from the default prior 0.5, should render a change in their inferred value detectable). We crossed all possibilities (0|0 or 1|1 as observed bigram, 0.2 or 0.8 for *p*) and generated 100 sequences for each (hence 400 sequences total). Second, at the end of each of these initial sequences, the prediction for the unobserved bigram, $p_{before}$, was queried by retrieving the output of the network after giving it as input '0' if the unobserved bigram was 0|0 or '1' otherwise. Third, the network was further presented with *m* repeated observations of the same value: '1' if the observed bigram was 1|1 or '0' otherwise. Finally, after this streak of repetition, the new prediction for the unobserved bigram, $p_{after}$, was queried (as before) and we measured its change with respect to the previous query, $|p_{after} - p_{before}|$. This procedure was repeated for *m* ranging from 2 and 75.

For statistics, we ran a one-tailed two independent samples t-test to test whether the mean change in unobserved bigram prediction of the population trained on coupled change points was higher than that of the population trained on independent change points.

## Complexity analyses

The complexity analysis reported in *Figure 8* consisted in measuring, for each network architecture and each environment, the performance of optimally trained networks as a function of the number of units N. For optimal training, hyperparameter optimization was repeated at several values of N, for each type of network and each environment (the resulting values are listed in *Table 1*). For the complexity analysis, a grid of equally spaced N values in logarithmic space between 1 and 45 was generated, an additional value of 474 was included specifically for the networks without recurrent weight training so as to match their number of trained parameters to that of an 11-unit gated recurrent network, and an additional value of 1,000 was included specifically for the networks without gating to facilitate the extrapolation. For every value on this grid, 20 networks of a given architecture in a given environment were randomly initialized with the set of hyperparameter values that was determined to be optimal for the nearest neighboring N value in logarithmic space. The performance of these networks after training was evaluated using a new couple of training and test datasets per environment, each consisting of 400 minibatches of 20 sequences for training and 1000 sequences for testing.

## Statistics

To assess the variability between different agent solutions, we trained 20 agents for each type of agent and each environment. These agents have different random seeds (which changes their parameter initialization and how their training data is shuffled). Throughout the article, we report mean or median over these agents, and individual data points when possible or 95% confidence intervals (abbreviated as "CI") otherwise, as fully described in the text and figure legends. No statistical methods were used to pre-determine sample sizes but our sample sizes are similar to those reported in previous publications (*Masse et al., 2019*; *Yang et al., 2019*). Data analysis was not performed blind to the conditions of the experiments. No data were excluded from the analyses. All statistical tests were two-tailed unless otherwise noted. The data distribution was assumed to be normal, but this was not formally tested. The specific details of each statistical analysis are reported directly in the text.

## Code availability

The code to reproduce exhaustively the analyses of this paper is available at https://github.com/cedricfoucault/networks_for_sequence_prediction and archived on Zenodo with DOI: 10.5281/zenodo.5707498. This code also enables to train new networks equipped with any number of units and generate *Figures 2–7* with those networks.

## Data availability

This paper presents no experimental data. All synthetic data are available in the code repository at https://github.com/cedricfoucault/networks_for_sequence_prediction and archived on Zenodo with DOI: 10.5281/zenodo.5707498.

# Acknowledgements

We thank Yair Lakretz for useful feedback, advice, and discussions throughout the project, Alexandre Pouget for his input when starting this project, and Charles Findling for comments on a previous version of the manuscript.

## Additional information

### Funding

| Funder | Grant reference number | Author |
|---|---|---|
| École normale supérieure Paris-Saclay | PhD fellowship "Contrat doctoral spécifique normalien" | Cédric Foucault |
| Agence Nationale de la Recherche | 18-CE37-0010-01 "CONFI LEARN" | Florent Meyniel |
| H2020 European Research Council | ERC StG 947105 "NEURAL PROB" | Florent Meyniel |

The funders had no role in study design, data collection and interpretation, or the decision to submit the work for publication.

### Author contributions

Cédric Foucault, Florent Meyniel, Conceptualization, Formal analysis, Funding acquisition, Methodology, Project administration, Supervision, Visualization, Writing - original draft, Writing - review and editing

### Author ORCIDs

Cédric Foucault  http://orcid.org/0000-0002-7247-6927
Florent Meyniel  http://orcid.org/0000-0002-6992-678X

### Decision letter and Author response

Decision letter https://doi.org/10.7554/eLife.71801.sa1
Author response https://doi.org/10.7554/eLife.71801.sa2

## Additional files

### Supplementary files

• Transparent reporting form

### Data availability

This paper presents no experimental data. All synthetic data are available in the code repository at https://github.com/cedricfoucault/networks_for_sequence_prediction and archived on Zenodo with https://doi.org/10.5281/zenodo.5707498.

The following dataset was generated:

| Author(s) | Year | Dataset title | Dataset URL | Database and Identifier |
|---|---|---|---|---|
| Foucault C | 2021 | Networks for sequence prediction | https://github.com/cedricfoucault/networks_for_sequence_prediction | Github, prediction |
| Foucault C | 2021 | Networks for sequence prediction | http://dx.doi.org/10.5281/zenodo.5707498 | Zenodo, 10.5281/zenodo.5707498 |

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
