## [Editor Report]

There has been a longstanding interest in developing normative models of how humans handle latent information in stochastic and volatile environments. This study examines recurrent neural network models trained on sequence-prediction tasks analogous to those used in human cognitive studies. The results demonstrate that such models lead to highly accurate predictions for challenging sequences in which the statistics are non-stationary and change at random times. These novel and remarkable results open up new avenues for cognitive modelling.

---

## [Decision Letter]

**Decision letter after peer review:**

Thank you for submitting your article "Gated recurrence enables simple and accurate sequence prediction in stochastic, changing, and structured environments" for consideration by *eLife*. Your article has been reviewed by 3 peer reviewers, including Srdjan Ostojic as the Reviewing Editor and Reviewer #1, and the evaluation has been overseen by Michael Frank as the Senior Editor. The following individual involved in review of your submission has agreed to reveal their identity: Mehrdad Jazayeri (Reviewer #2).

The three reviewers are enthusiastic about the manuscript, but have found that the main claims need to be contextualised or rephrased to avoid giving an overstated impression. In the absence of any direct comparison with human behavior and/or neural activity, and considering the high degree of abstraction in model (11 units with abstract computational building blocks), the paper needs a major revision in Discussion to highlight the gap between the results and neurobiology/behavior.

The Reviewing Editor has drafted a consolidated review to help you prepare a revised submission.

Essential revisions:

1. The most notable weakness of the paper is that is not clear whether its aim is to develop a neural model that is close to optimal or a neural model that explains how biological brains handle stochasticity and volatility. There is no serious and quantitative comparison to behavior or neural data recorded in humans or animal models. All the comparisons are with other algorithms and reduced GRU networks. One can appreciate these comparisons if the goal is to show that a full GRU network is close to optimal (which, in many cases, it is). But do humans exhibit a similar level of optimality? One possibility would have been to provide some sort of analysis that would show that the types of errors the model makes are in some counterintuitive (or even intuitive) way like the types of errors humans make. In some of the papers where certain heuristics were proposed, the entire goal was to explain characteristic sub-optimalities in human behavior. As an example, see the recent paper from the Koechlin group in Nature Human Behavior. More generally, there is no shortage of papers quantifying human behavior in stochastic volatile environments. It would be great to see that the errors humans make in at least some task map onto the errors GRU networks make. Imagine for example that such a comparison would show that human errors are more similar to a lesioned version of the GRU, even though the full GRU is closer to optimal. The natural conclusion for such an observation would be that some of the proposed mechanisms are in fact not at play. In any case, all the reviewers think the comparison to human behavior would be valuable, and should be at minimum extensively discussed.

2. On the importance of gating: a lot of emphasis is put on the necessity of gating (eg title, abstract, discussion line 478). But the methods used in the paper cannot demonstrate necessity. Indeed other studies (see eg Collins, Sohl-Dickenstein and Sussillo, arxiv 2016) have argued that gating in RNNs improves their trainability, but does not increase their capacity. That study argued that large vanilla RNNs are able to reach the same performance as gated RNNs with more extensive training and/or hyper-parameter tuning. The claims and discussion should be revised to reflect this limitation.

3. The biological relevance of gating seems also somewhat over-stated (eg in the abstract): while there is no doubt various forms of gating are present in the nervous system, how they map to the specific time-dependent form used in GRUs is far from clear. The relationship of these gate variables with actual synapses, neurons, or populations of neurons is at best speculative at this point.

4. In terms of comparing to biology, the discussion states that "mapping between artificial units and biological neurons may not be straightforward." But biological and artificial models can still be compared quite effectively in terms of activity in the state space, and these comparisons can help reject hypotheses quite effectively. Training RNNs have been a productive avenue for understanding neural computations in the past years, in many studies of this class networks are constrained or contrasted by experimental data (Mante and Sussillo et al., 2013, Rajan et al., 2016 or Finkelstein and Fontolan et al., 2021 as some examples). It could have been possible to try to understand the geometry of neural representations of latent variables in network dynamics and how it is learned and depends on the environment. Additionally, by performing dynamical system analysis (see eg Susillo and Barak, 2013 or Dubreuil and Valente et al., bioRxiv as examples) it might be possible to understand the role of gating in the network computations.

5. The focus on very small network does not necessarily seem relevant when comparing with biologic networks (the phrase "reasonably sized networks" on l.479 seems inappropriate). The analysis of network size in Figure 7 goes until 45 units, which remains very small, and it's difficult to extrapolate the results to larger networks. For instance, large vanilla RNNs implement an effective form of gating based on their non=linearity (Dubreuil et al. 2020), and this mechanism may be able to drastically increase sequence-prediction performance.

6. Another weakness of the paper is that, for each new task, it trains a new GRU. Humans seem to be able to adapt to changes in the latent structure of the generative process without massive retraining. How does this flexibility map onto the proposed scheme? In one of the supplements, cross-task performances have been shown. One notable result is that a GRU trained on a changing bigram with or without coupled change points does quite poorly on the changing unigram. This is an example of failed generalization from a much more complex latent structure to a simpler one, which is indicative of overfitting (to the structure of a generative model – not its parameters). Somewhat counterintuitively, for the GRU model (as well as various other models), the smallest hit on generalization performance occurs when the models are trained on the changing unigram, which is the simplest latent structure considered. This is consistent with several psychophysical studies suggesting that humans may not rely on accurate latent models and may instead rely on simpler heuristics. In the end, is it justified to train new GRUs for each task?

7. Note that LSTMs are able to perform similar computations like the ones in this study here as is shown in Wang and Kurt-Nelson et al., 2019.

8. As a more technical point, the comparison with networks without gating does not seem fully fair. Freezing gating effectively reduces the number of time-dependent variables by a factor 3. Also, when freezing gating, one could treat the gating parameters as fixed hyper-parameters to be optimized, rather than setting them by hand to one.

---

## [Author Response]

Essential revisions:1. The most notable weakness of the paper is that is not clear whether its aim is to develop a neural model that is close to optimal or a neural model that explains how biological brains handle stochasticity and volatility. There is no serious and quantitative comparison to behavior or neural data recorded in humans or animal models. All the comparisons are with other algorithms and reduced GRU networks. One can appreciate these comparisons if the goal is to show that a full GRU network is close to optimal (which, in many cases, it is). But do humans exhibit a similar level of optimality? One possibility would have been to provide some sort of analysis that would show that the types of errors the model makes are in some counterintuitive (or even intuitive) way like the types of errors humans make. In some of the papers where certain heuristics were proposed, the entire goal was to explain characteristic sub-optimalities in human behavior. As an example, see the recent paper from the Koechlin group in Nature Human Behavior. More generally, there is no shortage of papers quantifying human behavior in stochastic volatile environments. It would be great to see that the errors humans make in at least some task map onto the errors GRU networks make. Imagine for example that such a comparison would show that human errors are more similar to a lesioned version of the GRU, even though the full GRU is closer to optimal. The natural conclusion for such an observation would be that some of the proposed mechanisms are in fact not at play. In any case, all the reviewers think the comparison to human behavior would be valuable, and should be at minimum extensively discussed.

The primary aim of our study is to develop neural models that are both close to optimal and simple (i.e. with a small number of units), and to determine under what conditions they can do so, rather than to develop models that can be directly compared with biological brains. Still, the models we develop can inform neuroscience insofar as the tasks we have chosen are tasks that humans and other animals are capable of doing, and in which they show the specific qualitative aspects of optimality that we have investigated (even if they are otherwise suboptimal in several ways). We have modified the Introduction (l. 28, 30, 71) and the Abstract (l. 13) to make our goal clearer. We also now provide further details on several citations throughout the Results by pointing to the relevant figures of previous papers where these qualitative signatures are observed in humans (see l. 197–198, 241, 242–243, 406).

The direct comparison with the brain (behavioral or neural data), and in particular its suboptimalities, remains a very interesting future direction and it was not sufficiently discussed in the previous version of the manuscript. We have added a section in the Discussion dedicated to this topic and have incorporated new elements: see the section "Suboptimalities in human behavior" l. 607. In particular, we have detailed three possible ways to explore suboptimality with the networks: using networks with less training, using networks with fewer units or sparser connections, or using networks that are altered in some way (as suggested by the reviewers).

Note that although there is no shortage of experimental data on learning in stochastic and volatile environments in general, a direct comparison of the data between our study and previous experimental studies can rarely be made, either because the participant responses are categorical choices (often binary) rather than continuous estimates (e.g. Findling, Chopin, and Koechlin, 2021; Findling and Wyart et al. 2019), or because the generative process is very different (such as when observations are sampled from a Gaussian, e.g. Nassar et al. 2010; 2012; Prat-Carrabin et al., 2021). The lack of experimental data suitable for direct comparison is even more pronounced in the case of the changing bigram environments (the second and third environments in our study): the only data we are aware of are those collected in our lab, which have the shortcoming that participant responses are far too infrequent (one question every ~15 observations on average, Meyniel et al. 2015; 2017; 2019; 2020). We intend to acquire new data (including trial-by-trial estimates) to allow such a comparison in the future.

2. On the importance of gating: a lot of emphasis is put on the necessity of gating (eg title, abstract, discussion line 478). But the methods used in the paper cannot demonstrate necessity. Indeed other studies (see eg Collins, Sohl-Dickenstein and Sussillo, arxiv 2016) have argued that gating in RNNs improves their trainability, but does not increase their capacity. That study argued that large vanilla RNNs are able to reach the same performance as gated RNNs with more extensive training and/or hyper-parameter tuning. The claims and discussion should be revised to reflect this limitation.

We agree with the reviewer that our study cannot prove necessity in the strict mathematical sense. Proving necessity would require proving the non-existence of other architectures with similar performance; in practice we can only compare a limited number of architectures (one could conceive of others), and even within these architectures, we cannot test the infinity of possible parameter values. We had tried to say this in the Discussion paragraph about the minimal set of mechanisms but we now realize based on the reviews that it is not sufficient. We have rephrased this Discussion paragraph (see l. 544–560), and screened our text to eliminate phrasing suggestive of strict necessity (including in the Abstract, the Introduction, the Results, and the Discussion).

We also agree that a much larger vanilla RNN can achieve the same task performance as a smaller gated RNN. We intended to demonstrate this point through Figure 8 and the related text. To better convey this message, we have rephrased the text (see new paragraph l. 466 and legend l. 463), and have added to Figure 8 a new data point corresponding to a much larger number of units for the vanilla RNN, to facilitate the extrapolation and indicate that a larger vanilla RNN can ultimately approach optimality. We interpret this as evidence of the advantage afforded by gating to perform the computation simply, i.e. with few units (see also our response to comment #5).

This slow growth of the vanilla RNN’s performance with the number of units is well described by a power law. More precisely, if *N* is the number of units, and *p* is the % of optimal performance, the law would be: (100 – *p*) = c (1 / *N*)^α^. We fitted this law in the unigram environment with a least-squares linear regression on the logarithm of *N* and (100 – *p*) using the data points from 2 to 45 units, and obtained a goodness-of-fit R^2^=92.4%. We then extrapolated to *N*=1000 using the fitted parameters, and found that the predicted performance was within 0.2% of the performance we actually obtained for networks of this size (predicted: 97.8%, obtained: 97.6%), which further confirms the validity of the power law. Based on this power law, more than 10^4^ units would be needed for the vanilla RNN to reach the performance of the GRU at 11 units. We have reported this power law analysis in the revised manuscript (see new paragraph l. 466).

Regarding trainability: gating is best known indeed for improving the network’s trainability; however, that gating seems advantageous for performing the computation we’re interested in with few units, and not just for trainability, is one outcome of our study that we find interesting. We tried as much as possible to eliminate the trainability factor and approach the best possible performance for each network architecture by conducting an extensive hyperparameter optimization (repeated for each task, each architecture, and several numbers of units). One indication that this procedure worked well is that a plateau is reached (Snoek et al., 2012): the optimal value was always found in the first three quarters of the procedure (most often in the first half); in the last quarter, the validation performances of the new samples are almost identical (although lower), which contrasts with the highly variable performance of the first samples and indicates that Bayesian optimization does not gain from further exploration. Still, we have modified the text to mention the issue of trainability and better gauge the strength of the claim (see paragraph l. 556).

Our findings are not at odds with Collins et al.'s (2016) argument that gating does not increase the capacity of a RNN, because capacity (as measured in their study) is not what we measured. In their study, capacity was defined either as the number of bits per parameter that the RNN can store about its task during training, or as the number of bits per hidden unit that the RNN can remember about its input history. What we measured, and what we’re interested in, is the capability to perform the specific type of probabilistic inference in the specific type of environments that we have introduced (not to perform any task). In fact, capacity is actually what we want to control for rather than measure: given a certain memory capacity, does a particular architecture perform better than another? (See also our response to comment #5 about simplicity.)

3. The biological relevance of gating seems also somewhat over-stated (eg in the abstract): while there is no doubt various forms of gating are present in the nervous system, how they map to the specific time-dependent form used in GRUs is far from clear. The relationship of these gate variables with actual synapses, neurons, or populations of neurons is at best speculative at this point.

We fully agree and this is actually what we meant when we listed different possible candidates of gating in biology: it is speculative. We have strengthened this point by now stating it explicitly in the Discussion (see l. 564). What we meant was that since gating as a computational mechanism seems useful for solving the kind of problems that the brain faces, it is an invitation for us as neuroscientists to see if we can interpret the processes at play in the brain as doing gating, and it is all the more welcome given that, in biology, many forms of gating have already been observed. We also agree that the GRU has a very specific form of gating and we did not mean to imply that it is only this very specific form that one should consider. When exploring biological substrates it is therefore important not to be too attached to the precise form of gating of the GRU. We have rephrased the Discussion to stress this point (see l. 564–566) and have provided additional references for the possible biological implementations of gating (l. 573–574 and 574–576).

4. In terms of comparing to biology, the discussion states that "mapping between artificial units and biological neurons may not be straightforward." But biological and artificial models can still be compared quite effectively in terms of activity in the state space, and these comparisons can help reject hypotheses quite effectively. Training RNNs have been a productive avenue for understanding neural computations in the past years, in many studies of this class networks are constrained or contrasted by experimental data (Mante and Sussillo et al., 2013, Rajan et al., 2016 or Finkelstein and Fontolan et al., 2021 as some examples). It could have been possible to try to understand the geometry of neural representations of latent variables in network dynamics and how it is learned and depends on the environment. Additionally, by performing dynamical system analysis (see eg Susillo and Barak, 2013 or Dubreuil and Valente et al., bioRxiv as examples) it might be possible to understand the role of gating in the network computations.

First, concerning the comparison to biology, please see our response to comment #1.

Second, we would like to thank the reviewers for their suggestion which allowed us to illustrate our point in a different, geometrical and telling way. We have followed the reviewers’ suggestion and made a new figure (analogous to figure 2 and 5 in Mante and Sussillo et al., 2013) that illustrates the dynamics of network activity in the state space, with and without gating, and how these relate to the ideal observer behavior—see Figure 4b. This helps to understand the network computations and the difference that gating makes. The geometry of the trajectories shows that, with gating, the network is able to separate the information about the precision of its estimate from the information about the prediction and to use the former to adapt its rate of update in the latter, whereas without gating, these two are not separated.

This allowed us to see that, in the network without gating, the decoded precision seemed very strongly dependent on the prediction. To quantify this dependence, we computed the mutual information between the decoded precision and the network’s prediction. It turned out to be very high in the network without gating (median MI=5.2) compared to the network with gating (median MI=0.7) and the ideal observer (MI=0.6). Note that the mutual information is not zero in the ideal observer (and the GRU) because precision tends to be higher for more predictable observations (i.e. when the prediction gets closer to 0 or 1). This is consistent with the rest of our results and completes our argument because adaptive behavior leverages the part of precision that is independent of the prediction.

We have incorporated this supplementary analysis and the new figure into our results by splitting the old figure 4 and the corresponding section of the Results into two figures and sections, revamping the text and figures accordingly (see l. 251–295, l. 230, l. 296, Figure 4, and Figure 5), and completing the Methods (l. 870–877 and l. 866–867).

This suggestion also helped us to illustrate the perturbation experiment (see bottom left diagram in Figure 5).

5. The focus on very small network does not necessarily seem relevant when comparing with biologic networks (the phrase "reasonably sized networks" on l.479 seems inappropriate). The analysis of network size in Figure 7 goes until 45 units, which remains very small, and it's difficult to extrapolate the results to larger networks. For instance, large vanilla RNNs implement an effective form of gating based on their non=linearity (Dubreuil et al. 2020), and this mechanism may be able to drastically increase sequence-prediction performance.

Please see our response to comment #1 about our primary goal which is not to develop networks directly comparable with biological neural networks. The phrase "reasonably sized networks" was misleading in that respect and we removed it; thank you for pointing it out.

In response to comment #2, we have added a data point to Figure 8 to facilitate the extrapolation to larger vanilla RNNs.

As for the biological implementation of this gating, we quite agree that it remains an open question: do biological neural networks use a mechanism to perform this gating without many neurons, or do they use a very large number of neurons to perform an effective gating as a vanilla RNN would (these are not mutually exclusive)? We have added the latter to our list of possible biological implementations of gating, along with the references that detail how this effective form of gating can be achieved (Beiran, Dubreuil, Valente, Mastrogiuseppe, Ostojic, Neural Computation 2021; Dubreuil, Valente, Beiran, Mastrogiuseppe, Ostojic, bioRxiv) (l. 574–576).

Regarding our focus on small networks, it is motivated by the desideratum of *simplicity*, which has two advantages:

1) The reduced model description, which provides better understanding. As scientists, we do not merely want our model to perform the task, we also want to understand how it does it. Constraining the size of the network ensures that the algorithm it performs can be described simply, i.e. with a few effective state variables. Knowing which key computational building blocks enable such simple solutions provides insight into the functioning of the system. This is similar to model reduction approaches as described in (Jazayeri and Ostojic, 2021, last paragraph before the conclusion), such as the reduction to a 2-unit network in (Schaeffer et al., 2020), or the reduction to an effective circuit with 2 internal variables in (Dubreuil et al., 2020).

2) The efficiency of the solution (low-memory, low-computational complexity). This is relevant for the brain insofar as the brain's computational resources are limited (Lieder and Griffiths, 2020). Here by “computational resources” we mean more precisely the amount of memory required for the computation, which is often quantified by the Shannon capacity, i.e. the number of bits that can be transmitted per unit of time (see for example Bates and Jacobs 2020; Bhui, Lai, and Gershman, 2021). In our case, this amounts to the number of units (each unit stores the same number of bits, encoded by the hidden state). Therefore, the minimum number of units sufficient for near-optimal performance gives us a measure of efficiency. (Orhan and Ma, 2017) also used this measure of efficiency.

Given the reviewers’ comments, it seems that this point about simplicity was not sufficiently well conveyed in the previous version of the manuscript. We have modified the Introduction (paragraph l. 73) to better motivate our focus on small networks and relate it to simplicity more explicitly, and have further elaborated on it in the Discussion including the above two advantages (l. 548–555).

6. Another weakness of the paper is that, for each new task, it trains a new GRU. Humans seem to be able to adapt to changes in the latent structure of the generative process without massive retraining. How does this flexibility map onto the proposed scheme? In one of the supplements, cross-task performances have been shown. One notable result is that a GRU trained on a changing bigram with or without coupled change points does quite poorly on the changing unigram. This is an example of failed generalization from a much more complex latent structure to a simpler one, which is indicative of overfitting (to the structure of a generative model – not its parameters). Somewhat counterintuitively, for the GRU model (as well as various other models), the smallest hit on generalization performance occurs when the models are trained on the changing unigram, which is the simplest latent structure considered. This is consistent with several psychophysical studies suggesting that humans may not rely on accurate latent models and may instead rely on simpler heuristics. In the end, is it justified to train new GRUs for each task?

Regarding the cross-task performances, it seems that there was some misunderstanding because our results actually show the opposite: it is the GRU trained in the more complex environment (either of the bigram environments) that generalizes best to the simpler environment (unigram) (Figure 6—figure supplement 1). The reviewer's comment made us realize that this figure was difficult to read in the previous version. We therefore grouped the data differently and present of another set of comparisons to highlight this result more clearly: for one GRU trained in a given environment, the performances in the three test environments are now side by side, which allows the reader to better see the generalization performance given one training environment and to compare it with that given a different training environment (see Figure 6—figure supplement 1).

Regarding the question of whether it is justified to train a new GRU for each environment given that humans seem to be able to adapt to the environment without massive retraining: In fact, it would be unfair to compare the GRUs’ generalization performance as presented here with humans’ ability to generalize as observed in our lab, because when humans do this task in the lab, they are explicitly told what the latent structure is (i.e., the generative process of the observations), they do not have to discover it, unlike GRUs. This point was mentioned but not explicit enough, we now explain it in a new Discussion paragraph (l. 633).

In this study, we focused on the ability to leverage the latent structure during inference rather than the ability to discover this structure during training. From a theoretical point of view, neither the GRU nor humans can be expected to discover the structure purely from the observations without a large sample size, since even an ideal observer model that arbitrates between the two bigram structures in a statistically optimal fashion requires many observations to determine the correct structure—see Heilbron and Meyniel (2019) p.14:

“In our task, the optimal hierarchical model is able to correctly identify the current task structure (coupled vs. uncoupled change points), but only with moderate certainty even after observing the entire experiment presented to one subject (log-likelihood ratios range from 2 to 5 depending on subjects) [one experiment corresponds to 4 sequences i.e. 4*380=1520 observations]. […] We speculate that in real-life situations, some cues or priors inform subjects about the relevant dependencies in their environment; if true, then our experiment in which subjects were instructed about the correct task structure may have some ecological validity.”.

Regarding humans’ ability to flexibly switch from one structure to another without retraining given a cue about the current structure, it would be interesting to study the same ability in our network. This could be done by giving an additional input to the network that codes for the cue. We now mention this future direction in the new Discussion paragraph (see l. 637–641).

7. Note that LSTMs are able to perform similar computations like the ones in this study here as is shown in Wang and Kurt-Nelson et al., 2019.

Thank you for reminding us to mention the LSTM because it is a very popular architecture and many readers are likely to think about it too. We agree: the LSTM incorporates gating mechanisms similar to that of the GRU that allow it to perform the same computation. We have verified this in practice by repeating the hyperparameter optimization, training, and testing procedure with the LSTM: we indeed obtain a performance comparable to the GRU—see Author response image 1 (99% in the unigram environment and 98% in the bigram environment). We added a note in the paper to mention that the LSTM architecture also incorporates gating and can achieve the same performance as the GRU (l. 690–692) and have rephrased our exposition of the architectures to indicate that the GRU is only one particular case of a ‘gated recurrent’ architecture (see l. 136–137).

**Author response image 1. sa2fig1:** At an equal number of units, the LSTM matches the GRU in performance but is more complex.

The reason we had chosen the GRU over the LSTM is that we were looking for the minimal sufficient architecture, and the LSTM is a more complex architecture than the GRU, which turned out sufficient. LSTM units are more complex than GRU units in two ways: they have three gates instead of two, and they have an additional state variable called “cell state” (or “memory cell”) that adds to the hidden state. Thus, for the same number of units, the LSTM has not only more parameters than the GRU (at 11 units, the LSTM has 629 parameters and the GRU 475 parameters), but also and more importantly a state space twice as large as that of the GRU and the other architectures we considered (at 11 units, the number of state variables is 22 for the LSTM and 11 for the GRU and the others; see response to main comment #8 about which variables count as state variables). Besides, the introduction of the cell state means that we cannot always perform the same analyses and interventions that we perform on the other architectures.

8. As a more technical point, the comparison with networks without gating does not seem fully fair. Freezing gating effectively reduces the number of time-dependent variables by a factor 3. Also, when freezing gating, one could treat the gating parameters as fixed hyper-parameters to be optimized, rather than setting them by hand to one.

It seems that clarifying the definition of *variables* is key to answer this question. Removing gating does not reduce the number of *state variables* of the system because what we called the “gating variables” (*r* and *z*) are not state variables. The hidden state (*h*) is the only state variable since it alone suffices to determine the future behavior of the system (GRU and others). Our use of the gating variables is merely for convenience of exposition, to make the GRU more intelligible to us researchers (by labeling the factors in the equation that correspond to gating). One can equivalently characterize the system without these variables using a single recurrence equation that contains only the hidden state. We added a note to mention this (l. 683–684). Furthermore, note that when gating is removed, even when tripling the size of the state space, the vanilla RNN does not reach the performance of the GRU (Figure 8).

Regarding the possibility to treat the gating parameters as fixed hyper-parameters, it is an interesting possibility. In the case of *r*, if we’re not mistaken, it should not change anything because this fixed hyper-parameter could be absorbed into the recurrent weights (w’=rw), which are optimized during training. In the case of *z*, it would amount to treating the time constant of the units as a hyperparameter. We have added a sentence in the Methods to mention this possibility (l. 698).